# Drug and disease signature integration identifies synergistic combinations in glioblastoma

Vasileios Stathias[1,2,3,4], Anna M. Jermakowicz [1,2,3,4], Marie E. Maloof [1,2,4], Michele Forlin[3], Winston Walters[4], Robert K. Suter[2], Michael A. Durante [5], Sion L. Williams[2,6], J. William Harbour[5], Claude-Henry Volmar[1,2,4], Nicholas J. Lyons[7], Claes Wahlestedt[1,2,4], Regina M. Graham[2,4], Michael E. Ivan[2,4], Ricardo J. Komotar[2,4], Jann N. Sarkaria [8], Aravind Subramanian[7], Todd R. Golub [7,9,10,11], Stephan C. Schürer[1,2,3] & Nagi G. Ayad[1,2,4]

Glioblastoma (GBM) is the most common primary adult brain tumor. Despite extensive efforts, the median survival for GBM patients is approximately 14 months. GBM therapy could benefit greatly from patient-specific targeted therapies that maximize treatment efficacy. Here we report a platform termed SynergySeq to identify drug combinations for the treatment of GBM by integrating information from The Cancer Genome Atlas (TCGA) and the Library of Integrated Network-Based Cellular Signatures (LINCS). We identify differentially expressed genes in GBM samples and devise a consensus gene expression signature for each compound using LINCS L1000 transcriptional profiling data. The SynergySeq platform computes disease discordance and drug concordance to identify combinations of FDA-approved drugs that induce a synergistic response in GBM. Collectively, our studies demonstrate that combining disease-specific gene expression signatures with LINCS small molecule perturbagen-response signatures can identify preclinical combinations for GBM, which can potentially be tested in humans.

[1] Center for Therapeutic Innovation, Department of Psychiatry and Behavioral Sciences, University of Miami Miller School of Medicine, 1120 NW 14th St, Miami, FL 33136, USA. [2] Sylvester Comprehensive Cancer Center, University of Miami Miller School of Medicine, 1475 NW 12th Ave, Miami, FL 33136, USA. [3] Department of Molecular and Cellular Pharmacology, Center for Computational Science, University of Miami Miller School of Medicine, 1120 NW 14th St, Miami, FL 33136, USA. [4] University of Miami Brain Tumor Initiative, Department of Neurosurgery, University of Miami Miller School of Medicine, 1095 NW 14th Terrace, Miami, FL 33136, USA. [5] Bascom Palmer Eye Institute, Sylvester Comprehensive Cancer Center, Interdisciplinary Stem Cell Institute, University of Miami Miller School of Medicine, 900 NW 17th St, Miami, FL 33136, USA. [6] Department of Neurology, University of Miami Miller School of Medicine, 1150 NW 14th St, Miami, FL 33136, USA. [7] Broad Institute of Harvard and MIT, 415 Main St, Cambridge, MA 02142, USA. [8] Department of Radiation Oncology, Mayo Clinic, 200 First St SW, Rochester, MN 55902, USA. [9] Harvard Medical School, 25 Shattuck St, Boston, MA 02115, USA. [10] Howard Hughes Medical Institute, 4000 Jones Bridge Rd, Chevy Chase, MD 20815, USA. [11] Department of Pediatric Oncology, Dana-Farber Cancer Institute, 450 Brookline Ave, Boston, MA 02215, USA. These authors contributed equally: Vasileios Stathias, Anna M. Jermakowicz. Correspondence and requests for materials should be addressed to S.C.S. (email: sschuerer@med.miami.edu) or to N.G.A. (email: nayad@miami.edu)

Glioblastoma (GBM) is the deadliest form of brain cancer with a median two-year survival of 14% and a progression-free survival period of 6.9 months[1–5]. The current standard of care includes surgical resection followed by radiation and temozolomide (TMZ) administration. However, inherent or acquired resistance to both radiation and TMZ is nearly universal. Radiation-induced double-strand breaks (DSBs) can be overcome by genetic alterations such as the prevalent *EGFRvIII* amplification and TMZ-induced DNA base mispairs, which requires both a functioning mismatch repair (MMR) mechanism and a suppressed O6-methylguanine-methyltransferase (MGMT) activity[6]. As a result of the selective pressure that TMZ applies in a clinical setting, cells with abnormal MGMT expression and/or inactivation of MMR proteins gain a survival advantage and contribute to resistance to therapy[7,8].

This nearly universal resistance to ionizing radiation and TMZ treatment clinically has prompted many groups to search for novel targeted therapies for GBM[4]. Ideally, combination treatments should be identified to reduce the likelihood of resistance pathway upregulation after utilization of any one targeted therapy. For instance, studies have shown that combining bromodomain and extra-terminal (BET) domain protein inhibitors with other compounds may eliminate resistance mechanisms in multiple cancers[9–12]. However, identifying such combinations is a challenge in GBM given the intratumoral heterogeneity[13]. To overcome potential resistance to BET inhibitors in GBM, we developed a computational platform, SynergySeq, to identify compounds that can be used in synergistic combinations with a reference compound, such as a BET inhibitor (Fig. 1). The platform utilizes the extensive L1000 transcriptional-response profiles generated by the LINCS Project and creates perturbation-specific

transcriptional signatures, and subsequently integrates these drug signatures with disease-specific profiles derived from TCGA Consortium transcriptional data[14–16]. The LINCS perturbagen-response transcriptional profiles are generated using the L1000 assay, which is a high-throughput bead-based assay that measures the expression of 978 representative landmark transcripts[17]. Since the LINCS L1000 datasets lack GBM-specific transcriptional signatures, we treat GBM PDX and stem-like cells with the bromodomain inhibitor JQ1, and find that JQ1 inhibition of GBM cells yields a characteristic transcriptional signature. By comparing the differential gene expression changes induced by other compounds to the GBM-JQ1 transcriptional signature, we identify compounds that synergize with BET inhibitors in reducing GBM cell expansion in vitro and in vivo. Importantly, we demonstrate that our platform, which was originally developed for BET inhibitor combinations in GBM, can be utilized to identify novel FDA-approved drug combinations. Collectively, our studies provide a novel platform, SynergySeq, which can identify patient-specific drug combinations in GBM.

## Results

**The L1000 genes cluster different cancer types**. To evaluate whether the levels of the 978 transcripts that are measured by the L1000 assay can be utilized to distinguish among the different transcriptional landscapes of The Cancer Genome Atlas (TCGA) cancer types, we extracted the 978 L1000 genes from TCGA RNA-Seq data. Overall, 4515 TCGA RNA-Seq samples were downloaded belonging to the following cancer types: 546 uterine corpus endometrial carcinoma (UCEC) samples, 166 rectum adenocarcinoma (READ) samples, 156 GBM samples, 1097 breast invasive carcinoma (BRCA) samples, 479 colon adenocarcinoma

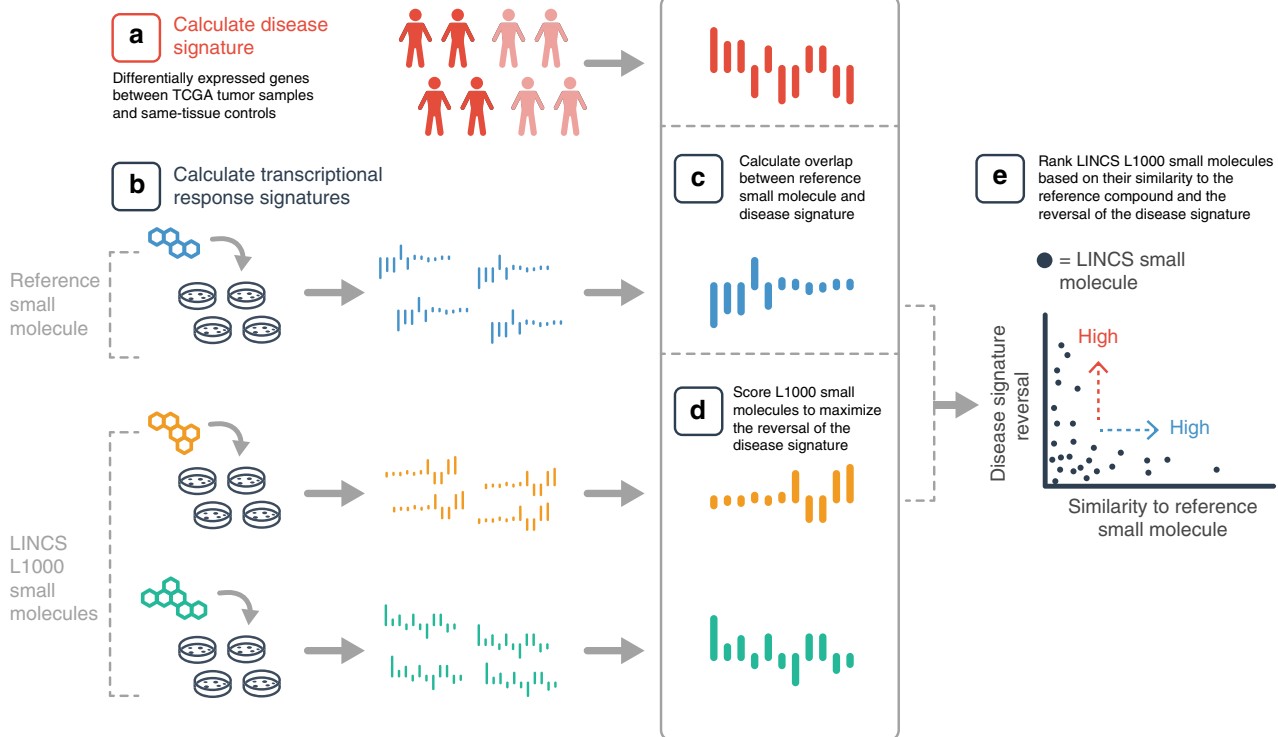

**Fig. 1** SynergySeq workflow for identifying synergistic drug combinations using disease discordance and drug concordance. **a** A disease signature is calculated by identifying the differentially expressed genes between tumor samples and same-tissue controls. **b** Transcriptional consensus signatures (TCS) are calculated for a reference small molecule and the LINCS L1000 small molecules. **c** The overlap between the reference TCS and the disease signature is calculated. **d** The LINCS L1000 small molecules are ranked to maximize the reversal of the disease signature. **e** The LINCS L1000 small molecules are plotted based on their similarity to the reference small molecule and the reversal of the disease signature

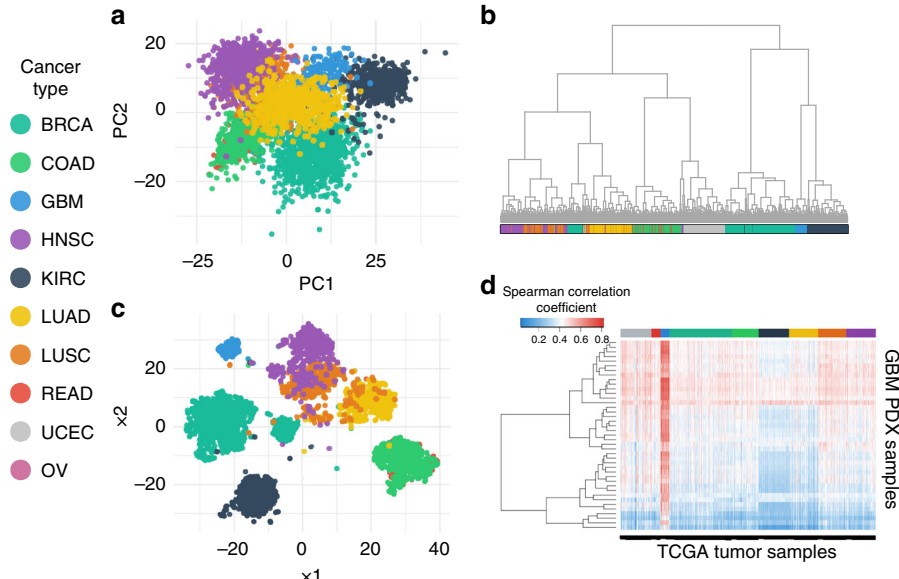

**Fig. 2** Clustering of TCGA samples and GBM PDX samples using the L1000 gene set reflects cancer types. **a** PCA plot, **b** Hierarchical Clustering, and **c** t-distributed stochastic neighbor embedding (tSNE) plot of 4515 RNA-Seq TCGA tumor samples labeled based on their cancer type by using the expression of only 978 genes. **d** Heatmap of the Spearman correlation of 41 PDX GBM samples from Mayo Clinic and 4515 tumor samples downloaded from TCGA. Cancer types: BRCA: breast invasive carcinoma, COAD: colon adenocarcinoma, GBM: glioblastoma, HNSC: head-neck squamous cell carcinoma, KIRC: kidney renal clear cell carcinoma, LUAD: lung adenocarcinoma, LUSC: lung squamous cell carcinoma, READ: rectum adenocarcinoma, UCEC: uterine corpus endometrial carcinoma, OV: ovarian cancer

(COAD) samples, 533 kidney renal clear cell carcinoma (KIRC) samples, 515 lung adenocarcinoma (LUAD) samples, 503 lung squamous cell carcinoma (LUSC) samples, and 530 head and neck squamous cell carcinoma (HNSC) samples.

Figure 2 shows clustering of cancer types using principal component analysis (PCA) (Fig. 2a) and t-distributed stochastic neighbor embedding (tSNE) (Fig. 2c) with the individual tumor samples colored by their respective tumor-type. Of interest is the overlap of the lung cancers (LUSC and LUAD), as well as overlap of the colorectal cancers (COAD and READ), demonstrating that samples coming from similar tissue types have a similar gene expression profile and that this transcriptional similarity can be revealed by comparing the expression levels of the subset of 978 L1000 genes.

Furthermore, hierarchical clustering was performed using the expression of the 978 genes of the same 4515 RNA-Seq samples (Fig. 2b). Again, each cancer type has been labeled with a different color and we can see that the 978 L1000 genes are sufficient to cluster the GBM samples together.

**Integrating the Brain Tumor TCGA and L1000 datasets**. We hypothesized that the L1000 perturbation-response transcriptional profiles would sufficiently describe the impact of small molecule treatments on the cancer cell transcriptome. These profiles could then be used to identify small molecule combinations that maximize the efficacy for GBM and other cancer types.

To determine whether we could utilize patient-derived xenograft (PDX) samples to identify combinations for GBM based on the L1000 assay, it was necessary to evaluate whether the PDX samples are indeed transcriptionally representative of directly isolated human GBM tumor samples. RNA-Seq data of 41 GBM PDX samples obtained from the Brain Tumor PDX national resource at the Mayo Clinic were compared to the above 4515 TCGA RNA-Seq samples. The Spearman correlation coefficient (SCC) between the PDX and TCGA samples was then calculated and the results were plotted in a heatmap. As seen in

Fig. 2d, the TCGA samples with the highest SCC are the GBM samples, suggesting that cells derived from the Mayo Clinic GBM PDXs are transcriptionally similar to the GBM tumor samples in TCGA and are not similar to other tumors.

The LINCS L1000 datasets contain transcriptional profiles of more than 30 cell lines treated with more than 1700 small molecules. However, even though most cancer types are represented by multiple cell lines, the dataset does not contain any GBM cell lines. We therefore screened our own GBM samples using the L1000 platform. Two PDX GBM cell lines and four GBM stem-like cell lines were treated with a selection of compounds and the RNA was isolated and processed via the L1000 assay.

**Compound-specific transcriptional consensus signatures**. Since the majority of the LINCS L1000 transcriptional profiles are from non-GBM cells, it was necessary to create a new type of transcriptional signature that would be both independent of the cell type and representative of the perturbagen used. For this, we aggregated all the L1000 data and calculated the respective transcriptional consensus signature (TCS) for each compound. Using the plate-normalized Level 4 L1000 data, we identified compounds that had been used in multiple cell lines and for each compound we calculated a gene expression signature based on the genes that were consistently up or down regulated regardless of the cell line used. This core set of genes would be indicative of the transcriptional responses that each compound treatment induced.

Since the number of genes in each TCS varied greatly from compound to compound (Fig. 3a), we wanted to further evaluate this difference. We downloaded the transcriptional activity scores (TAS) for 197 compounds in the LINCS L1000 December 2015 dataset, calculated the median TAS for each compound, and then compared the median TAS to the total number of genes that had a non-negative TCS score. TAS is a metric that quantifies a compound's ability to induce an L1000 transcriptional response[17].

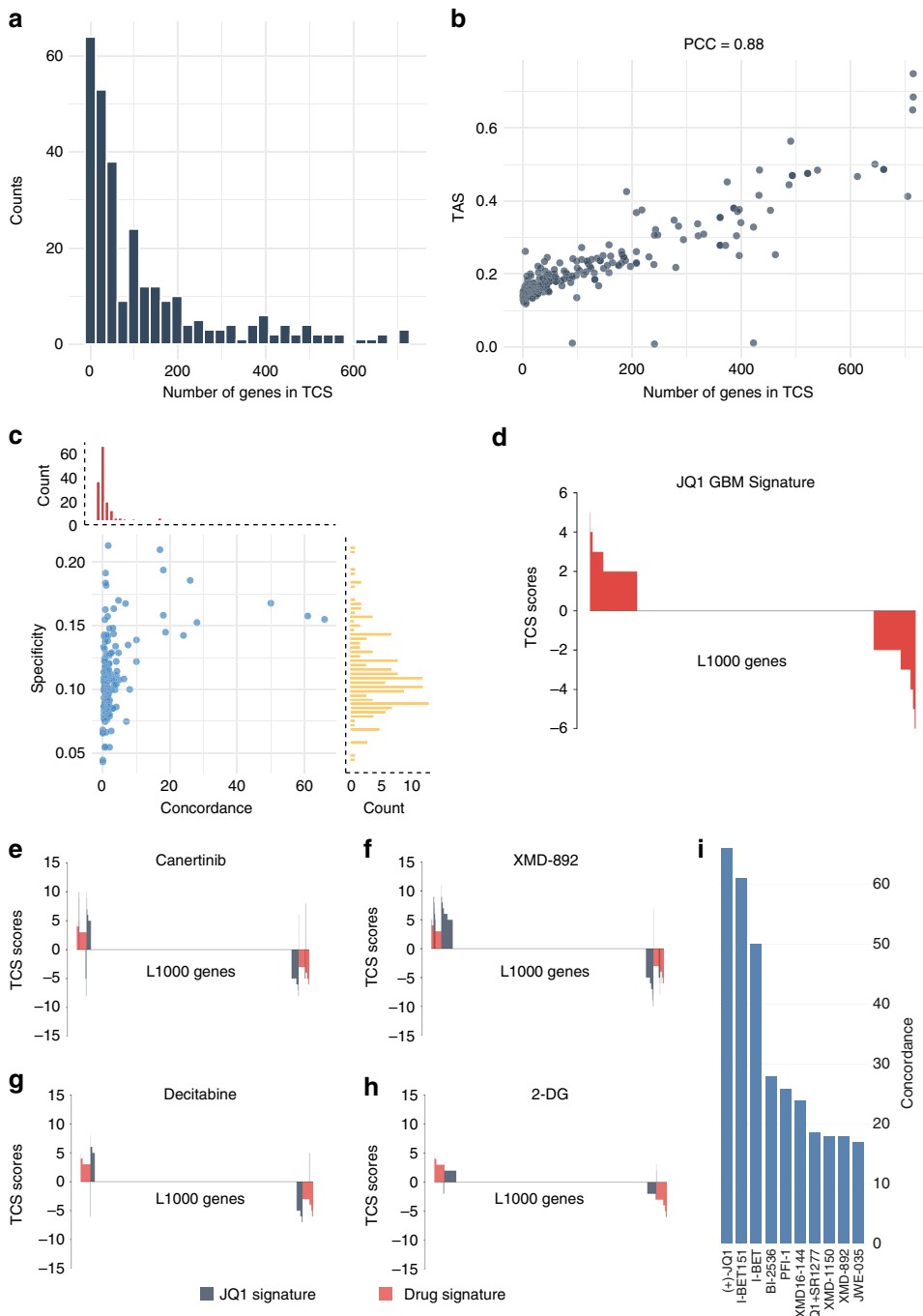

**Fig. 3** Transcriptional consensus signatures can identify common transcriptional responses to compounds. **a** Histogram of the number of genes in the TCS per compound. Most of the compounds tested have a low number of genes in their TCS. **b** The number of genes that each compound has in its TCS is correlated with the Transcription activity score, a metric for the consistency in a compound's transcriptional response. PCC: Pearson correlation coefficient. **c** Specificity and concordance of L1000 compounds compared to the JQ1 TCS. Only a few compounds overlap and have a concordant transcriptional response with the JQ1 TCS. **d** The JQ1 Transcriptional response signature. The top two (**e**, **f**) and the bottom two (**g**, **h**) compounds with the highest specificity to the JQ1 TCS. **i** The top 10 compounds with the highest concordance to the JQ1 TCS are BET inhibitors

Figure 3b shows that compounds with more genes as part of their TCS have a higher TAS, indicating an overall stronger and more consistent L1000 transcriptional response (Pearson correlation = 0.88, Spearman correlation = 0.85). Similarly, compounds with fewer genes in their TCS have a lower TAS and a weak L1000 transcriptional response.

We then created a TCS using our own L1000 GBM data to evaluate if the general perturbational signatures were indicative of a response in GBM cells. The GBM-JQ1 signature was generated; it consisted of 264 genes that had a non-zero TCS score as shown in Fig. 3c. A higher TCS gene score (max score = 6 since six GBM cell lines were used) indicates that the gene was consistently over/under-expressed across different GBM cell lines. To obtain a robust GBM-JQ1 reference signature, we only considered genes that were concordantly over/under-expressed in at least half of our 6 GBM cell lines (TCS score ≥3 or ≤−3). This led us to a high confidence signature of 84 genes that were over- or underexpressed consistently after JQ1

treatment in at least half of our GBM cell lines (Supplementary Table 1).

We wanted to evaluate the transcriptional differences between the GBM-JQ1 signature and the TCSs generated from the LINCS L1000 dataset. As we sought compounds that would have a notable overlap with the 84-gene signature of our JQ1 compound, we excluded compounds with a low number of genes in their TCS (fewer than 50% of the genes in the GBM-JQ1 signature). We then calculated the specificity and the concordance of those compounds when compared to the GBM-JQ1 signature. Specificity (S) was defined as the percentage of genes that are common between a compound's TCS and the GBM-JQ1 TCS while normalizing for the total number of genes in the TCS (Eq. 1). In order to evaluate not only the overlap but also the directionality of the transcriptional changes, we utilized the concordance ratio. The concordance ratio (CR) was defined as the ratio of a compound's genes that have the same direction as the GBM-JQ1 signature and those that have the opposite direction (Eq. 2):

$$S = \frac{\sum_{i=1}^{978}[a_i \cdot b_i]}{\sum_{i=1}^{978}[a_i]} \text{ with } a_i = \begin{cases} 1, & \text{if} \quad z_i \neq 0 \\ 0, & \text{if} \quad z_i = 0 \end{cases} \text{ and}$$
$$b_i = \begin{cases} 1, & \text{if} \quad r_i \neq 0 \\ 0, & \text{if} \quad r_i = 0 \end{cases} \quad (1)$$

$$CR = \frac{\sum_{i=1}^{978}[a_i]}{\sum_{i=1}^{978}[b_i]} \text{ with } a_i = \begin{cases} 1, & \text{if} \quad z_i \cdot r_i > 0 \\ 0, & \text{if} \quad z_i \cdot r_i < 0 \end{cases} \text{ and}$$
$$b_i = \begin{cases} 1, & \text{if} \quad z_i \cdot r_i < 0 \\ 0, & \text{if} \quad z_i \cdot r_i > 0 \end{cases} \quad (2)$$

where z and r are the TCS vectors of the compound and GBM-JQ1, respectively.

Figure 3c shows the specificity plotted against the concordance of each compound. Compounds with high specificity induced transcriptional changes in genes that are also changed after JQ1 treatment. To further illustrate this overlap, we plotted the top 2 compounds and the bottom 2 compounds against the GBM-JQ1 TCS (Fig. 3e–h, respectively). Importantly, high specificity does not necessarily indicate that the compound induces gene expression changes in the same direction as JQ1. Rather, the concordance ratio considers the directionality of the induced transcriptional changes of a compound, making it a more useful metric. As seen in the histogram of Fig. 3i, most compounds with a variable overlap with the GBM-JQ1 signature have a low

concordance ratio. However, compounds with a high concordance ratio also show high pharmacological similarity to JQ1, highlighting the importance of not only taking into account the correct genes but also the directionality of the transcriptional changes.

From this analysis, we were able to conclude that the GBM-JQ1 signature, despite having fewer genes, was highly concordant with the overall LINCS transcriptional consensus JQ1 signature, thus validating that the transcriptional response induced by JQ1 in GBM cells is highly similar to the response in other cells. In addition to the compound-specific pharmacological relevance, we found that TCS, in contrast to the average replicate signatures, significantly reduced cell-specific bias thus improving the integration and comparability of the GBM PDX L1000 data with the LINCS L1000 reference data (Supplementary Figure 1).

**TCSs characterize drug classes by mechanism of action.** Using TCSs corresponding to small molecules used in the LINCS L1000 and the PDX GBM datasets, we calculated the Pearson correlation of all compound pairs and performed hierarchical clustering and consensus clustering (Fig. 4a, Supplementary Figure 2a). Plotting compounds with high transcriptional similarities (Pearson Correlation >0.7) also showed an MOA-specific grouping (Fig. 4b).

In all cases, one of the most prominent clusters was for the bromodomain inhibitors, which included the BET inhibitor, JQ1[18]. This finding is consistent with the earlier observation of high similarity between the JQ1 TCSs of the LINCS reference and GBM datasets. Furthermore, both the BROAD and the Glioblastoma JQ1 TCSs clustered together with other known bromodomain inhibitors, as seen in the cluster labeled BET in Fig. 4a, b and Supplementary Figure 2a. These BET inhibitors included I-BET, I-BET151, and PFI-1. We also found several perturbagens in the BET inhibitor clusters that were not explicitly annotated as BET inhibitors. Upon an in-depth literature search, these compounds were determined to be dual inhibitors, which inhibit BET proteins and other targets simultaneously (Supplementary Figure 3). More specifically, BI-2536 is a PLK/BRD4 inhibitor[19], TG101348 is a dual JAK2/BRD4 inhibitor[20] and XMD11-50 is a dual LRRK2/BRD4 inhibitor[20].

In addition to the BET inhibitor cluster, multiple MOA classes such as the HSP inhibitor, MEK inhibitor, mTOR/PI3K inhibitor, and HDAC inhibitor compounds were also tightly clustered, supporting the idea that compounds of similar mechanisms of action produce similar transcriptional effects on cells and that TCSs robustly characterize such drug-induced gene expression

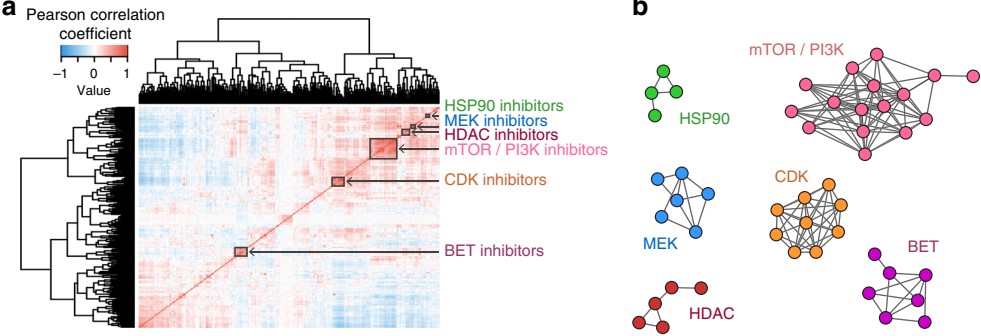

**Fig. 4** L1000 consensus signatures reflect molecule mechanism of action. **a** Correlation matrix of the 285 LINCS consensus signatures and the 14 GBM consensus signatures. The LINCS L1000 dataset was merged with the GBM L1000 dataset, and TCSs were calculated and used to produce the correlation matrix. The red clusters along the diagonal indicate compounds that have highly correlated consensus signatures. **b** Networks of highly correlated LINCS small molecules. A connection indicates a 0.7 or above Pearson correlation of their consensus gene signatures. The compounds were colored and labeled based on their mechanism of action as identified through a literature search. Compound names and mechanisms of action can be found in Supplementary Figure 3

**Table 1 Top 10 upregulated (top) and top 10 downregulated (bottom) L1000 genes with expression levels from mechanism of action networks in Fig. 4**

| BET | | mTOR/PI3K | | CDK | | HDAC | | HSP90 | | MEK | |
|---|---|---|---|---|---|---|---|---|---|---|---|
| PNKP | 9.29 | NPDC1 | 9.69 | PAK1 | 11.67 | PRCP | 10.4 | SMNDC1 | 10.75 | HSPB1 | 10 |
| SLC25A4 | 9 | MAP2K5 | 9.88 | CAMSAP2 | 11.78 | MNAT1 | 10.6 | SPAG7 | 11.25 | HMOX1 | 8.33 |
| CPNE3 | 9.14 | FOXO4 | 9.44 | TBXA2R | 11.89 | PTPN1 | 10.4 | NUDT9 | 11.5 | ECH1 | 7.5 |
| STUB1 | 8.71 | MBTPS1 | 9.63 | MAPK9 | 12.56 | FOXO4 | 10.4 | HSPD1 | 11.75 | TSC22D3 | 7.17 |
| AKR7A2 | 9.14 | HSD17B11 | 9.38 | TCEA2 | 12.56 | TCEA2 | 10.4 | SNX6 | 11 | NFIL3 | 9.17 |
| BNIP3L | 8.57 | CBLB | 9.38 | SOCS2 | 11.56 | IER3 | 11.2 | HSPA8 | 12 | PIK3R3 | 7.17 |
| STXBP1 | 10.29 | CDK19 | 9.75 | NFKBIE | 12 | RRAGA | 9.8 | KIAA0494 | 12 | PPIC | 7.83 |
| CIRBP | 8.14 | SNCA | 11.19 | ADAT1 | 12.11 | LIPA | 10.8 | ATP2C1 | 12 | NNT | 8 |
| TMEM2 | 8.29 | KLHDC2 | 11.25 | RAB27A | 11.89 | ANO10 | 10 | PRUNE | 11.75 | NFKBIA | 7.17 |
| LRPAP1 | 8.29 | POLD4 | 11.06 | GPC1 | 12.89 | SSBP2 | 9.8 | ARID4B | 11 | BRP44 | 8.17 |
| DECR1 | −7.29 | CYCS | −11.38 | COPB2 | −10.89 | PUF60 | −11.8 | SLC37A4 | −11.75 | STX1A | −9.83 |
| ATF1 | −6.86 | GTF2A2 | −11.38 | CDC25B | −11.11 | SUV39H1 | −13.4 | TSEN2 | −10.75 | DUSP6 | −10.67 |
| TMEM109 | −8 | BIRC5 | −10.06 | KIAA0528 | −12 | ADI1 | −13 | CETN3 | −10.75 | DUSP4 | −8.83 |
| CRYZ | −7 | ICMT | −9.94 | PCM1 | −11.33 | TIMELESS | −11.6 | TIPARP | −11.25 | EGR1 | −10.67 |
| IKBKE | −7 | MYBL2 | −9.88 | ATF1 | −11.67 | PFKL | −11.6 | NUP88 | −11.75 | CCND1 | −8.83 |
| ICAM3 | −7.86 | PGAM1 | −10.69 | CASC3 | −11.67 | MICALL1 | −11.6 | PXMP2 | −11.25 | SPRED2 | −8.33 |
| PYCR1 | −9.86 | MYCBP | −10.19 | SCAND1 | −11 | NUP85 | −11.8 | IER3 | −11 | FOSL1 | −9.5 |
| SORBS3 | −8 | PHGDH | −10.13 | ADAM10 | −12.22 | AKAP8 | −11.6 | PAFAH1B3 | −12 | ITGB5 | −10.33 |
| PSMB8 | −7.43 | RUVBL1 | −9.88 | CLTC | −11.78 | DCTD | −11.6 | CCDC85B | −13.25 | MAT2A | −7.17 |
| PLOD3 | −8.86 | POP4 | −11.13 | ATMIN | −11.56 | KEAP1 | −13.4 | AMDHD2 | −12.25 | HMGA2 | −8.5 |

changes (Table 1, Supplementary Figure 2, Supplementary Figure 3, Supplementary Data 3, Supplementary Data 4).

**L1000 TCSs Identify Compounds Targeting Orthogonal Pathways**. The nearly universal resistance to the standard TMZ treatment regimen in GBM, as well as other single-agent targeted therapies, suggests that targeting a single pathway may be insufficient to overcome the dysregulated and compensatory oncogenic signaling network[21,22]. Combination therapy is one approach to target the compensatory survival and proliferation pathways potentiated by a single agent and thus reduce the number of proliferating and resistant cancer cells.

By generating a TCS for each of the LINCS L1000 compounds, we can robustly characterize the transcriptional effect that each compound has on cancer cells and quantify the pairwise similarities between those treatments. Our hypothesis was that orthogonal TCSs would be predictive of efficacious compound combinations. As a proof of concept, we used the transcriptional signatures to identify a compound signature that is most dissimilar to the BET inhibitor JQ1, which has been shown to reduce proliferation of several cancer cells including GBM[23]. JQ1 binds to the acetyl-lysine binding site of BET-family bromodomains, shows high potency and specificity towards the BET bromodomain proteins BRD2, BRD3, BRD4, and BRDT and exhibits anti-proliferative effects in many cancer types[18,24–26].

Using the TCGA RNA-Seq dataset we identified 132 genes out of the 978 L1000 genes that were differentially expressed in the TCGA Glioblastoma samples relative to TCGA same tissue controls. Since the ideal compound would fully reverse this 132-gene glioblastoma disease signature, we calculated a disease-specific discordance ratio (DR) relative to JQ1 for all compounds. The disease discordance ratio was defined as the ratio of drug-induced differentially expressed genes that have the opposite direction to the glioblastoma disease signature and those that have the same direction, and which are absent from the GBM-JQ1 reference signature (Eq. 3). This method would rank highly compounds that target JQ1-orthogonal sets of genes that are part of the transcriptional signature of the disease of interest (glioblastoma):

$$\text{DR} = \frac{\sum_{i=1}^{978}[b_i \cdot c_i]}{\sum_{i=1}^{978}[a_i \cdot c_i]} \text{ with } a_i = \left\{ \begin{array}{ll} 1, & \text{if } z_i \cdot d_i > 0 \\ 0, & \text{if } z_i \cdot d_i < 0 \end{array} \right\},$$

$$b_i = \left\{ \begin{array}{ll} 1, & \text{if } z_i \cdot d_i < 0 \\ 0, & \text{if } z_i \cdot d_i > 0 \end{array} \right\}, \text{ and } c_i = \left\{ \begin{array}{ll} 1, & \text{if } r_i = 0 \\ 0, & \text{if } r_i \neq 0 \end{array} \right\} \quad (3)$$

where z, d, and r are the TCS vectors of the compound, the disease, and the reference (GBM-JQ1) signature, respectively.

Using the TCSs and the list of the 132 glioblastoma DEGs, we ranked the LINCS compounds according to their concordance to the reference GBM-JQ1 signature. The GBM-JQ1 synergy plot displays the TCGA GBM Discordance Ratio (Eq. 3) on the y-axis as a function of the JQ1 Concordance Ratio (Eq. 2) on the x-axis. A high x-axis value indicates that the compound's TCS is highly similar to GBM-JQ1. As shown in Fig. 5a, the compounds with the highest x-axis value are known BET inhibitors (JQ1, IBET, IBET-151). GBM-JQ1 denotes the JQ1 transcriptional signature generated using the GBM L1000 dataset and JQ1 denotes the JQ1 TCS generated using the L1000 LINCS reference dataset.

Combining the two dimensions of the synergy plot, namely the disease-specific discordance ratio and the concordance ratio to the reference (GBM-JQ1), each compound can be scored by a single value that quantifies its orthogonality to the transcriptional effect induced by JQ1. This TCS based Orthogonality Score (OS) is defined as the distance of each compound from the most concordant (highest x-axis value) and least discordant (lowest y-axis value) reference signature compound, in our case the GBM-JQ1 signature. After unity-based normalization (scaling into the range [0 and 1]) of CR and DR, OS is defined in Eq. 4.

$$\text{OS} = \sqrt{(1 - \text{CR})^2 + (\text{DR})^2} \quad (4)$$

**JQ1 synergizes with aurora inhibitors in vitro and in vivo**. We identified the Aurora kinase B inhibitor GSK-1070916 as the most orthogonal compound to JQ1 in the glioblastoma gene expression background (Fig. 5a)[27]. This compound has the highest DR value among all other compounds, suggesting that it had the most genes that were discordant compared to the GBM DEGs. GSK-

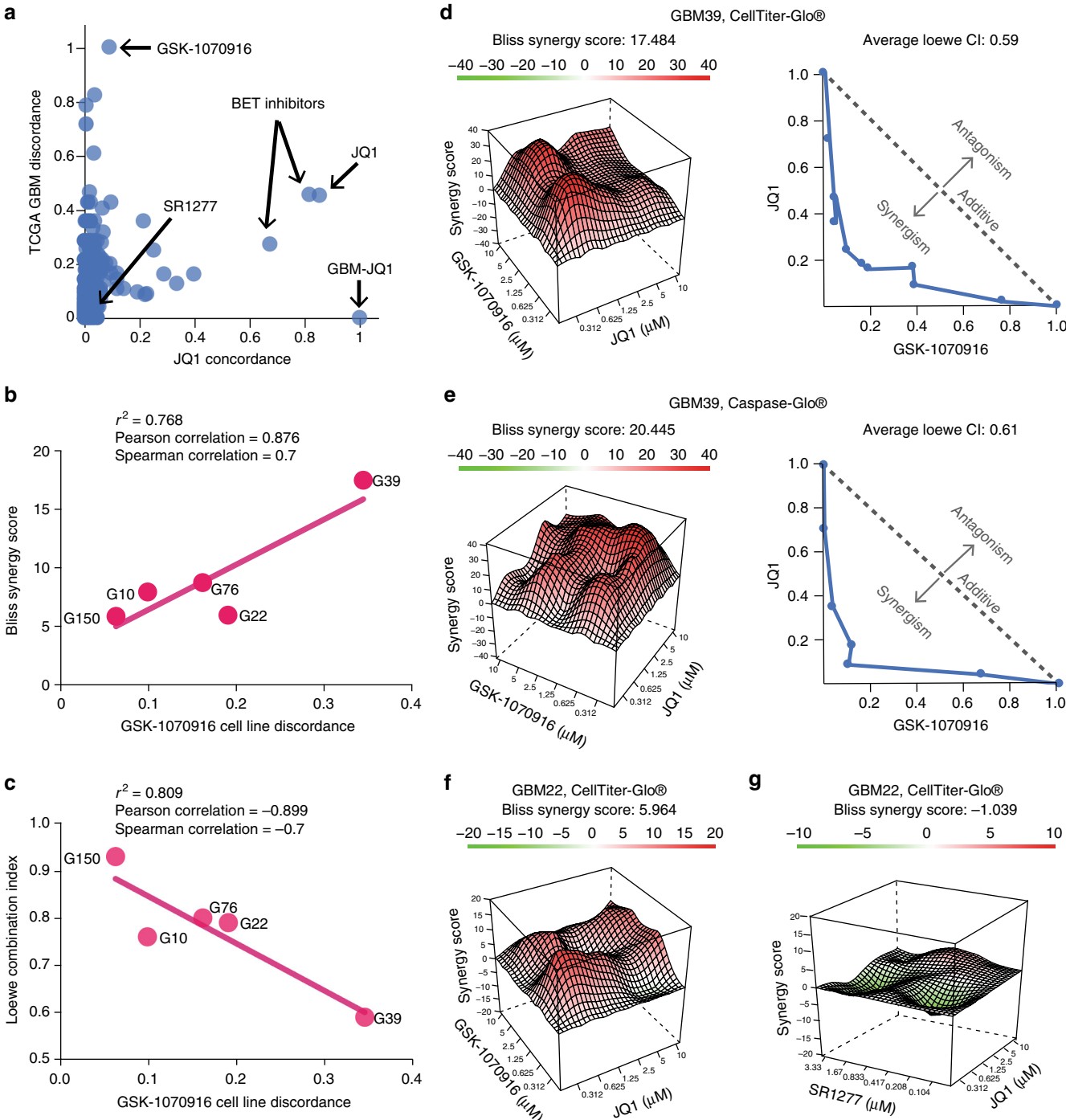

**Fig. 5** Synergistic response of cell proliferation inhibition and apoptosis after treatment with JQ1 and/or GSK-1070916. **a** Ranking of the 285 LINCS compounds based on their orthogonality to the GBM-JQ1 consensus signature. Compounds with a high x-axis value have a signature concordant to JQ1 and compounds with a high y-axis value have a signature discordant to the disease. **b**, **c** Synergy was assessed for a total of 5 cell lines and the Bliss synergy scores (**b**) and the Loewe combination indices (**c**) were plotted against the cell line-specific discordance from GSK-1070916. A strong correlation was seen between increased discordance and increased synergy (higher Bliss score, lower Loewe CI). **d** Reduced cell proliferation measured by ATP levels using CellTiter-Glo® are normalized to positive control (Velcade) and negative control (DMSO). From left to right, the Bliss score surface and an isobologram plot of the Loewe combination index for the combination of JQ1 with GSK-1070916. Synergy analyses for additional cell lines can be found in Supplementary Figure 4. **e** Apoptosis measured by Caspase3/7 levels using Caspase-Glo® and normalized to positive control (Velcade) and negative control (DMSO). **f** Synergistic response of the Bliss score surface observed using JQ1 with GSK-1070916 as measured by CellTiter-Glo®. **g** Sub-synergistic response of the Bliss score surface observed using JQ1 with SR1277 as measured by CellTiter-Glo®. Source data can be found in the Source Data file, Supplementary Data 6, under tab Fig. 5

1070916 also has a very low CR value, indicating that its transcriptional signature is not similar to the JQ1 reference signature, and the highest OS as the overall most transcriptionally orthogonal to GBM-JQ1.

To test our original hypothesis that compound combinations with orthogonal TCSs (i.e., high OS) would be efficacious, we assessed the combination of JQ1 with GSK-1070916 in in vitro assays using 6 PDX GBM lines that were obtained from the Brain Tumor Patient-Derived Xenograft National Resource at the Mayo Clinic. Sensitivity to the JQ1 and/or GSK-1070916 treatment was assessed using the CellTiter-Glo® assay, which measures the levels of the cellular ATP. A $7 \times 7$ combination matrix was created that contained the two compounds at different concentration ratios. A total of 4 matrices were fitted in a standard 384-well plate and were used to treat the PDX GBM cells.

To assess the synergy of the two compound treatments, two analyses were performed: a Bliss independence model combination matrix surface evaluation and a Loewe Additivity combination index isobologram analysis. The Bliss model, a widely utilized model in synergism studies, is based on the principle that compounds act independently and do not interfere with each other[28]. Bliss plots were generated using the SynergyFinder web application [https://synergyfinder.fimm.fi][29]. The isobologram plot is used to evaluate the Loewe additivity model of synergism by displaying the concentration of compound required to achieve a specific effect level in monotherapy divided by the concentration required in combination[30]. Points overlaying the diagonal of the plot indicate additivity, points below the diagonal indicate synergism between the compounds, and points above the diagonal indicate antagonism between the compounds.

To assess the synergistic effect in a patient-specific manner, we calculated the cell-line-specific discordance for GSK-1070916 in combination with JQ1 using the RNA-Seq data for five of the PDX cell lines. As seen in Fig. 5b, c, there is a strong correlation between the cell line-specific GSK-1070916 discordance ratio (DR as defined in Eq. 3) and the synergy metric (increased Bliss score or decreased Loewe combination index indicate increased synergy) and, as shown in Fig. 5d and Supplementary Figure 4, synergy is observed under both the Bliss independence model and the Loewe additivity model.

We also assessed apoptosis induction using the Caspase-Glo® assay, which measures the levels of Caspases 3 and 7 (Fig. 5e). These results complemented the CellTiter-Glo® assays and indicated that JQ1 and GSK-1070916 act synergistically and induce a higher level of apoptosis compared to each single agent alone.

As a negative control, we identified SR1277, a casein kinase 1δ inhibitor, as a compound with both a low CR value and a low DR value, and thus predicted to have a sub-synergistic effect in combination with JQ1[31]. To assess the effect of the combination, we screened matrices of SR1277 or GSK-1070916 with JQ1 in the same cell line. We found that the combination of GSK-1070916 and JQ1 produced a synergistic effect (Fig. 5f) and the combination of SR1277 and JQ1 produced a sub-synergistic effect (Fig. 5g).

To determine whether the predicted combination of Aurora kinase and Bromodomain inhibitors reduces tumor growth in vivo, we combined JQ1 with the Aurora kinase inhibitor alisertib. We chose alisertib as it is in clinical trials for multiple cancers while GSK-1070916 is no longer in clinical trials. In addition, both alisertib and GSK-1070916 have similar TCSs (Supplementary Figure 2a, b, Supplementary Data 5) indicating a similar transcriptional response profile. Furthermore, alisertib and GSK-1070916 have been shown to inhibit Aurora kinase A and Aurora kinase B in vitro and in vivo in multiple settings[32]. As seen in Fig. 6 and Supplementary Figure 4, JQ1 synergized with

alisertib in vitro, and a combination of alisertib and JQ1 reduced tumor growth of GBM22 cells in vivo more than either treatment alone. In addition, mice treated with the JQ1+alisertib combination did not have reduced weight relative to mice treated with control vehicle.

**Orthogonal FDA-approved drugs synergize in vitro**. As a test of our SynergySeq platform, we constructed a library of 197 FDA-approved compounds, of which 83 compounds have perturbational profiles determined by L1000, to identify FDA-approved compounds that may be beneficial in the treatment of GBM. Drugs were tested at a concentration of 1 μM in six PDX GBM cell lines. Gemcitabine was identified as a lead compound based on its consistent response in the cell viability assay and was therefore used in the following synergy studies.

Using the TCS method, compounds were clustered based on similarity of the transcriptional profile they induce in all cancer cells, including the GBM PDX cells (Fig. 7a). Compounds were further prioritized based on the effect on cell proliferation as measured by cellular ATP, as well as the calculated concordance to gemcitabine and discordance to the GBM disease signature. Two compounds were selected to test in synergy studies with gemcitabine as a proof of concept: mitoxantrone and imatinib. Both compounds were predicted to be highly orthogonal to gemcitabine, with a concordance score of 0.00 for imatinib and 0.012 for mitoxantrone on a scale of least to most concordant with 0 being the least concordant (Fig. 7b). A full report of the concordance ratio, discordance ratio, and Orthogonality Score can be found in Supplementary Data 1.

Cell line-specific discordance was calculated for the combinations of gemcitabine with mitoxantrone and imatinib and the discordance with the cell line was found to be positively correlated with reducing cell proliferation in vitro in our FDA screen (Fig. 7c, d). PDX GBM76 was selected to test for proof of concept since all three compounds were found to be effective in this line, based both on our initial drug screen and on the cell line-specific discordance score, which is necessary to provide robust synergy calculations[33]. The compound combinations were tested in a synergy matrix, as described previously, to evaluate the synergy potential. The results of the synergy data were visualized using the Loewe independence model, which demonstrated that the combination with the more orthogonal compound produced a more synergistic response (Fig. 7e, f). The three synergy measures (GBM76 discordance, Orthogonality Score, and Loewe combination index) were compared to the drug concordance to gemcitabine and an association was observed between the in vitro synergistic response and the predicted synergy metrics (Fig. 7b).

## Discussion

We developed SynergySeq, a novel platform for identifying synergistic combinations in GBM, which integrates baseline disease transcriptional data with perturbagen-induced transcriptional signatures to identify compound combinations predicted to be effective in a given cell line. We demonstrated that a 978-gene expression set (L1000) is sufficient to cluster cancer samples from same tissue of origin. We utilized the L1000 dataset to generate a TCS for each of the LINCS compounds, which was incorporated into subnetworks reflecting their mechanism of action. Various compound classes such as HDAC inhibitors, MEK inhibitors, and BET inhibitors clustered independently based on the TCS. We utilized SynergySeq to identify the compound that was most discordant to the GBM disease signature and orthogonal to JQ1, GSK-1070916, an Aurora kinase B/C inhibitor. We demonstrated that combining GSK-1070916 with JQ1 induces synergy in reducing proliferation of GBM cells and that another Aurora

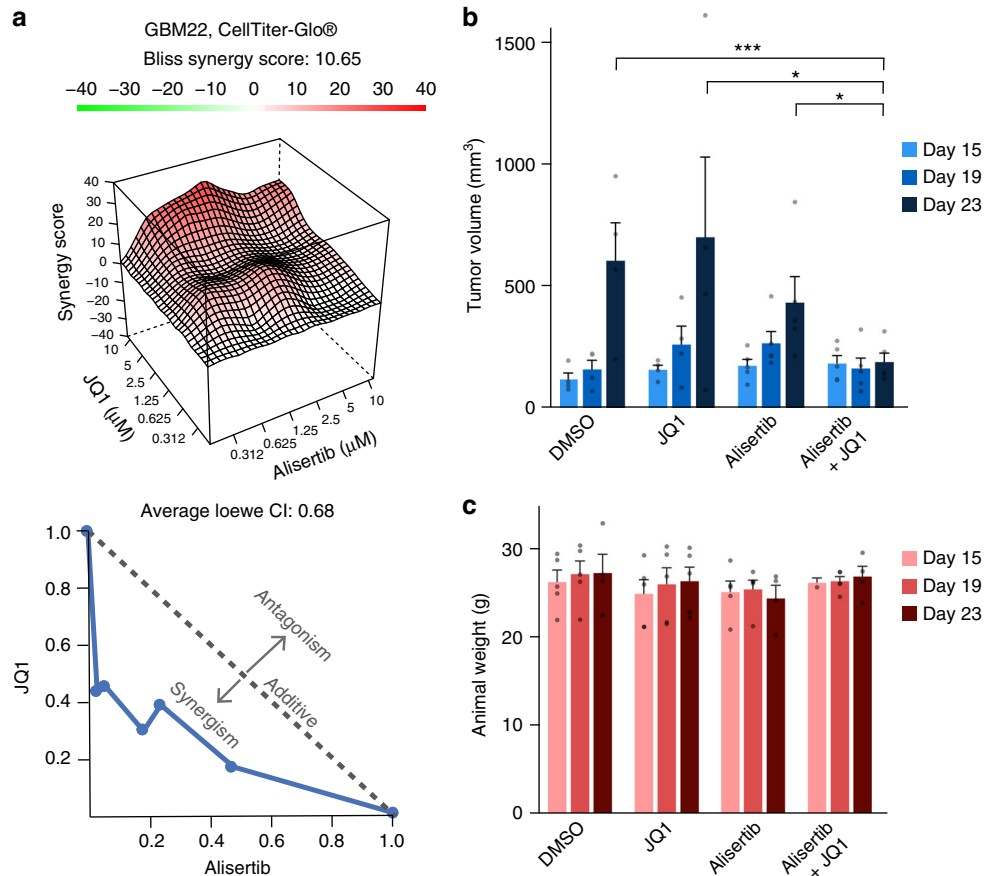

**Fig. 6** JQ1 and alisertib synergize in vitro and in vivo in reducing GBM tumor growth. **a** Synergistic response of JQ1 and alisertib measured by ATP levels using CellTiter-Glo® in GBM22 PDX cells. ATP levels are normalized to positive control (Velcade) and negative control (DMSO). The Bliss score surface (top) and isobologram plot of the Loewe combination index (bottom). **b** Synergistic response observed for JQ1 and alisertib combination in vitro in a flank model of GBM, measured by caliper. Mice were implanted with GBM22 PDX cells and treated starting at day 15 with 25 mg kg$^{-1}$ per day of alisertib and/or 30 mg kg$^{-1}$ per day of JQ1 for 10 days. Significance was determined using an unpaired $t$-test with Holm-Sidak correction for multiple comparisons, *$p <$ 0.05, ***$p <$ 0.001. **c** JQ1 and alisertib combination does not reduce mouse weight. Weight of mice analyzed in (**c**) was assessed at the indicated time points. Sample size was as follows: DMSO, JQ1 $n =$ 4; Alisertib, Alisertib+JQ1 $n =$ 5. Please note that one mouse in each the Alisertib and Alisertib+JQ1 group perished before 23-day animal weight was obtained. Mean and standard error bars are shown. Source data can be found in the Source Data file, Supplementary Data 6, under tab Fig. 6

kinase inhibitor, alisertib, synergizes with JQ1 to reduce tumor growth in vitro and in vivo. We also tested the platform for identifying FDA-approved compound combinations and demonstrate that gemcitabine and mitoxantrone or imatinib synergize in reducing GBM cell proliferation. The synergy detected by both cell proliferation and apoptosis assays was consistent with the predictions of the platform and suggests that the integration of the LINCS and TCGA transcriptional data can be a valuable tool in the identification of synergistic combinations in GBM.

The clustering of small molecules according to their transcriptional impact on cancer cells can be a valuable drug discovery and repurposing tool, with potential to shed light on a compound's mechanism of action (Supplementary Figure 3). GBM tumors consist of a highly heterogeneous cell population with varying gene signatures as well as mosaic expression for various GBM drivers[34]. Given the diverse cell population in GBM tumors, knowledge of the precise mechanisms of action and the transcriptional pathways targeted by compounds is critical in predicting compound efficacy in both monotherapy and combination therapy in a patient-specific manner. Furthermore, targeted therapies activate alternative cell survival pathways,

indicating the need for combination therapy to overcome resistance mechanisms[35–40].

A recent study demonstrated that compounds that either are or are not orthogonal can induce synergy in reducing cell proliferation or inducing apoptosis[41]. Our findings suggest that discordant disease signatures are also essential for achieving such synergy. For instance, the casein kinase 1δ inhibitor SR1277 was not concordant to the bromodomain inhibitor JQ1. However, as it was not discordant to the disease signature compared to JQ1, no synergy was observed when the two compounds were combined (Fig. 5g, h).

Using our SynergySeq platform, we calculated the cell line-specific discordance for a set of FDA-approved compounds. We were able to preselect gemcitabine, a pyrimidine nucleoside analog used to treat other solid tumors, which is a candidate for brain tumors based on its chemical properties[42,43]. To use in combination with gemcitabine, we selected mitoxantrone and imatinib based on the SynergySeq Orthogonality Score analysis and on our preliminary screening data. Mitoxantrone, an intercalating agent that inhibits topoisomerase II, has shown poor blood-brain barrier (BBB) penetrance but has improved survival when directly administered to tumors in vivo[44]. Imatinib, a

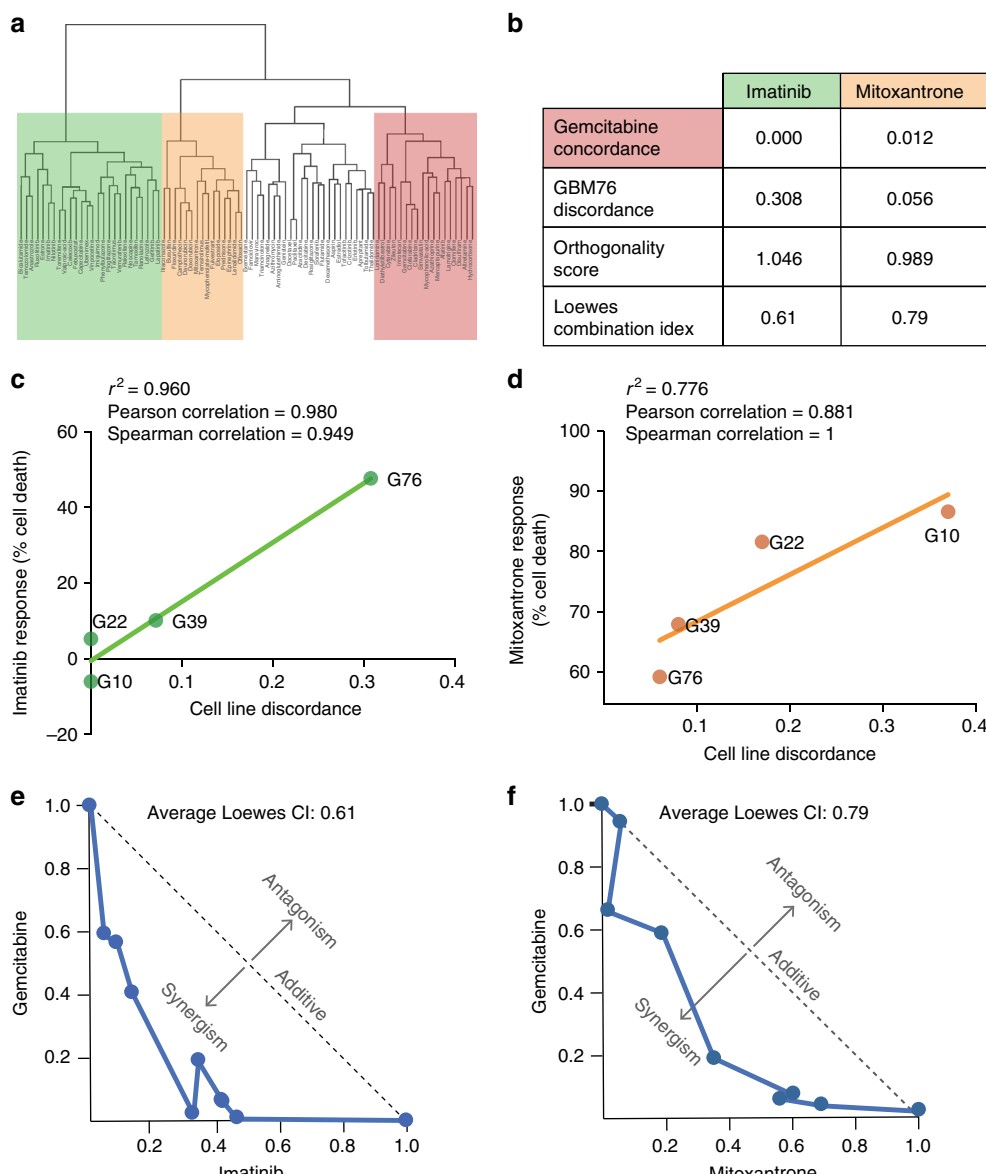

**Fig. 7** Combining FDA-approved compounds induces synergy in reducing GBM cell proliferation. L1000 profiling was performed for 83 FDA-approved compounds. **a** Compounds were clustered according to the transcriptional profile they induce in all cancer cells. Group 1, shaded in green, contains imatinib; Group 2, shaded in orange, contains mitoxantrone, and Group 3, shaded in red, contains gemcitabine. **b** Compounds were ranked based on concordance to gemcitabine and discordance from signature of the mean of the PDX GBM cell lines and two compounds were selected to test for proof of concept: imatinib and mitoxantrone. Orthogonality score was computed and the Loewes combination index was calculated in PDX GBM76 cells. **c**, **d** The reduced cell proliferation responses to imatinib and mitoxantrone in monotherapy for 4 cell lines were plotted against the cell line-specific discordance scores for imatinib (**c**) or mitoxantrone (**d**). **e**, **f** Combinations were tested in PDX GBM76 cells. Synergy was calculated using the Loewe additive model and a normalized isobologram was plotted to visualize synergy for each combination. Source data can be found in the Source Data file, Supplementary Data 6, under tab Fig. 7

tyrosine kinase inhibitor, has been investigated in a phase II clinical trial for GBM with limited efficacy although it was well tolerated when administered orally[45]. Importantly, we found that the calculated cell line-specific discordance of a compound was strongly correlated with the percentage of reduced cell proliferation seen for each compound in monotherapy (Fig. 7c, d). Furthermore, an increase in synergy metrics for drug concordance, cell line discordance, and Orthogonality Score in a specific cell line for two combinations were associated with an increase in the synergy seen in vitro (Fig. 7b). These data suggest that previously collected baseline gene expression data in a cell

line can be integrated with our L1000 compound TCS scores to generate patient-specific synergy predictions (Supplementary Figure 5).

By utilizing transcriptional data from LINCS, TCGA, and the Brain Tumor PDX national resource, we created compound-specific transcriptional signatures that could be used to identify combinations for GBM treatment. The biological relevance of those combinations was validated by both the use of external databases and the experimental validation using various in vitro assays. The SynergySeq platform is freely accessible at [http://synergyseq.com], where users may upload their normalized gene

expression data or use one of the pre-loaded TCGA cancer datasets. We anticipate that use of the application will allow investigators to identify possible drug combinations that can be tested in vitro and in vivo in preclinical models.

Never has the ability to predict drug efficacy in animal or human models been more important for the treatment of GBM. Not only is our armamentarium of candidate compounds increasing, but our capabilities of delivering them to the correct target are also improving. The BBB has always been a significant barrier to medical management of GBM, but novel less-invasive tools, such as laser interstitial thermal therapy and focused ultrasound, are improving our ability to penetrate tumors in patients[46,47]. SynergySeq could thus provide a unique resource not only by allowing investigators to preselect compounds with known BBB penetrance, but also by guiding the selection of non-brain penetrant drugs that can potentially cross the BBB using these novel tools. Furthermore, as we have generated a transcriptional signature for TMZ with radiation in GBM cells using L1000 data (Supplementary Table 1), investigators can potentially assess the synergistic potential of their investigational drugs with the current standard of care for GBM patients, although we acknowledge that one limitation of our approach is that TMZ induces a weak transcriptional signature. Another limitation is that most of the studies in LINCS utilized few compound concentrations and therefore input data into our platform can be improved by varying compound concentrations and possibly treatment times. Indeed, varying treatment conditions such as concentration can yield different transcription consensus signatures. However, overcoming these limitations could prove invaluable in the design of studies and clinical trials utilizing combination therapies for the treatment of brain tumors.

Previously, transcriptional profiling was insufficient to identify drug targets in other highly heterogeneous cancer subtypes[48]. However, reanalysis of these datasets by our platform can identify patient-specific and cancer type-specific drug combinations (Supplementary Figure 5 and Supplementary Figure 1). Importantly, analysis of previous datasets such as the NCI ALMANAC using our platform suggests that disease discordance is predictive of extent of synergy (Supplementary Figures 7–10).

Future studies will analyze whether SynergySeq can integrate single-cell sequencing data to identify drug combinations that target the cells responsible for tumor formation in a given GBM subtype. We found that single cell sequencing of GBM tumors identified a population of cells that contained the GBM disease signature as well as the JQ1, GSK-1070916, and alisertib signatures (Supplementary Figure 11). Several studies have shown that both GSK-1070916 and alisertib target both Aurora kinase A and Aurora kinase B[32]. Indeed, our single sequencing data demonstrate that Aurora kinase A and Aurora kinase B are expressed in the same population of cells (Supplementary Figure 11). These studies suggest that the synergy we observe may be due to modulating distinct pathways in a subpopulation of cells within the tumor rather than targeting two distinct subpopulations of cells. Further studies are required to test this directly in animals bearing tumors that will be treated with single agents and combinations to perform single-cell sequencing of remaining tumors. Importantly, our studies have demonstrated that the SynergySeq platform has the potential to predict effective drug combinations for preclinical in GBM and other cancers in a patient-specific manner.

## Methods

**Cell culture**. PDX GBM cells were obtained from the Brain Tumor PDX national resource at the Mayo Clinic. Cells were cultured in complete media consisting of Dulbecco's Modified Eagle's medium (DMEM):F12 with 10% fetal bovine serum and 1% penicillin and streptomycin (penn/strep). Cells were maintained for a maximum of 30 days before being discarded[49].

GBM stem-like cells were generated from patient tumor samples at the University of Miami using a previously described method and informed consent was obtained for all patients[50]. Cells were culture in media consisting of DMEM:F12 with 1% penicillin and streptomycin, 20 ng/ml each of human epidermal growth factor and human fibroblast growth factor, and 2% Gem21 NeuroPlex Serum-Free Supplement to form neurospheres.

**L1000 profiling of GBM samples**. GBM PDX cells or stem-like cells were treated with 10 μM JQ1, 100 μM SR1277, 10 μM JQ1 with 100 μM SR1277, 50 μM TMZ, 8 Gy ionizing radiation (IR) or 50 μM TMZ with 8 Gy IR and incubated for 24 h. Subsequently, cells were washed one time in phosphate buffered saline (PBS) and RNA was harvested using RNeasy Mini kit (Qiagen, Valencia, CA, USA) as per the manufacturer's instructions. RNA concentration was determined using a Nanodrop 2000 spectrophotometer (Thermo Scientific, Waltham, MA, USA) and 1 μg of RNA for a minimum of 2 biological replicates was sent for L1000 profiling. Two technical replicates were performed for each sample.

**Drug Screens**. PDX cells were plated in 25 μl of complete media in Nunc® 384-Well Tissue Culture Plates (Thermo Scientific, Waltham, MA, USA) at a concentration of 1000 cells per well or 3000 cells per well for the FDA-approved compound screen or the synergy screens, respectively. Cells were incubated overnight to establish adherent cultures, treated with 5 μl of drug dissolved in DMSO at a concentration of 10 μM and diluted with Hank's Balanced Salt Solution to six times the desired treatment concentration, and then incubated for 72 h. Finally, ATP content was measured using the CellTiter-Glo® Luminescent Cell Viability Assay (Promega Corporation, Madison, WI, USA) or the Caspase Glo® 3/7 Assay (Promega Corporation, Madison, WI, USA) following the manufacturer's protocol and plates were read on an EnVision Multilabel Plate Reader (Perkin Elmer). The FDA-approved compound screen consisted of two replicates each of 197 compounds at a concentration of 1 μM, DMSO as a negative control, and 10 μM Velcade as a positive control. Synergy screens consisted of a minimum of three replicates of 7 × 7 dose-response matrices, ranging from 10–0.3125 μM at 1:2 dilutions with seven replicates each of DMSO as a negative control and of 10 μM Velcade as a positive control. Final DMSO concentration was maintained at 0.1% throughout the experiments. In all screens, reduced cell proliferation was measured by normalizing the raw fluorescent values to the negative control (DMSO, 0% reduction) and the positive control (Velcade, 100% reduction) using the following formula:

$$\% \, reduced \, proliferation = 100 \times \left( \frac{LO - EC_0}{EC_{100} - EC_0} \right)$$

where LO is the raw luminescent output value, $EC_0$ is the mean raw luminescent of the negative control, and $EC_{100}$ is the mean raw luminescent output of the positive control.

Results of the drug screens can be found in Supplementary Data 2.

**Data integration**. A data processing pipeline was constructed to parse data from various databases, including TCGA, LINCS, and small molecule annotation databases. First, TCGA RNA-Seq datasets for GBM tumors and controls were downloaded, consolidated, and analyzed. Subsequently, LINCS data were downloaded and annotated. Transcriptional profiles were consolidated and stringent compound annotations were integrated from various databases to facilitate rapid data analysis.

**TCGA data**. The BROAD Institute's Firehose [https://gdac.broadinstitute.org] was used to download 153 TCGA RNA-Seq GBM tumor samples and 5 TCGA same tissue healthy controls. The RNASeqRawData function of the TCGA Assembler R package was also used to download 4515 Level 3 RNA Expression samples corresponding to 9 cancer types[51]. The R package DESeq2 was used to process and compare the raw counts of the TCGA RNA-Seq samples[52].

**LINCS data and L1000 dataset**. LINCS Data are generated by the LINCS DSGCs (Data Signature Generation Centers) and are made accessible for download by the LINCS Data Coordination and Integration Center (DCIC) via the LINCS Data Portal[53]. We developed a data processing pipeline consisting of modular scripts built using BIOVIA Pipeline Pilot (16.2) to parse the data and metadata from the DSGC databases via APIs and direct data access to files located in SFTP servers. An additional preprocessing step was performed for all gct and gctx files (generated by BroadT LINCS and PCCSE) by using the parse.gctx function of the cmapR R package.

Level 4 plate-normalized data were downloaded from the LINCS Data Portal for the L1000 December 2015 [http://lincsportal.ccs.miami.edu/datasets/#/view/LDS-1293] and March 2017 [http://lincsportal.ccs.miami.edu/datasets/#/view/LDS-1372] datasets. The processed dataset packages including the original data and metadata are available via the LINCS Data Portal [http://lincsportal.ccs.miami.edu/].

**Calculating consensus signatures**. The L1000 Level 4 datasets was filtered for only 24-hour treatment samples. Gene expression profiles were aggregated for samples using both the same small molecule and the same cell line (technical/biological replicates and/or use different doses of the small molecule). Aggregation was performed by first counting the number of samples that have a $|z\text{-score}| > 1$ for a particular gene. If this count was more than 20% of the total number of samples, then that particular gene was included in the aggregated expression profile.

Next, the above-aggregated gene expression profiles are collapsed at the small molecule level. Gene expression profiles that correspond to the same small molecule across all cell lines are aggregated to produce the TCS. A gene is included in the final TCS if it is up/down-regulated by a $|z\text{-score}| > 1$ in more than 30% of the cell lines that were treated by the same small molecule.

**Patient-derived xenograft GBM data**. Patient-derived xenograft (PDX) GBM tumor RNA-Seq data was obtained from the Brain Tumor PDX national resource at the Mayo Clinic. We downloaded the RPKM values, extracted the subset of values for the 978 L1000 genes, and normalized using the $\log_2$ fold change of the median RPKM count for each gene across all PDX GBM cell lines. The Mayo Clinic PDX GBM RNA-Seq data is available at the Mendeley Data repository [https://doi.org/10.17632/yz8m28gj6r.1].

**JQ1 and alisertib treatment in vivo**. GBM22 cells were implanted into the flanks of immunocompromised mice as previously described[25]. When the tumors reached 200 mm³ mice were injected with either DMSO, JQ1, alisertib, or JQ1+alisertib. Mice were injected twice daily intraperitoneally. Tumor growth was measured by caliper. Weight of mice was assessed at the indicated times. There were at least three experimental replicates for statistical analyses and investigators who conducted analysis were masked to the treatment groups. All animal experiments were approved by the IACUC at the University of Miami.

**SynergySeq platform**. The SynergySeq platform was developed using R Shiny [https://shiny.rstudio.com] and is freely available for use on the website: [http://synergyseq.com].

**Code availability**. The custom code required to generate these results is available at the Mendeley Data repository [https://doi.org/10.17632/yz8m28gj6r.1] and [https://github.com/schurerlab/SynergySeq].

## Data availability
Data supporting the findings of this study have been deposited at the Mendeley Data repository [https://doi.org/10.17632/yz8m28gj6r.1]. The L1000 datasets are available for L1000 December 2015 [http://lincsportal.ccs.miami.edu/datasets/#/view/LDS-1293], March 2017 [http://lincsportal.ccs.miami.edu/datasets/#/view/LDS-1372]. The Mayo Clinic PDX RNA-Seq data is available at the Mendeley repository [https://doi.org/10.17632/yz8m28gj6r.1]. The source data underlying Figs. 5b–g, 6a, b, 7c–f, and Supplementary Figures 4–6 are provided as a Source data file labeled Supplementary Data 6. A reporting summary for this Article is available as a Supplementary Information file. All other data supporting the findings of this study are available from the corresponding author, Dr. Nagi Ayad, upon request.

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

## Acknowledgements

We acknowledge funding from the NIH via grants R56102590 to N.G.A., U54CA189205 and U24TR002278 (Illuminating the Druggable Genome Knowledge Management Center, IDG-KMC, and Resource Dissemination and Outreach Center, IDG-RDOC) and U54HL127624 (BD2K LINCS Data Coordination and Integration Center, DCIC) to S.C.S. The Illuminating the Druggable Genome (IDG), Library of integrated Network-based Cellular Signatures (LINCS) and Big Data to Knowledge (BD2K) programs are funded by the trans-NIH Common Fund. IDG-KMC, IDG-RDOC and BD2K-LINCS DCIC are administered by the NCI, NCATS, and NHLBI, respectively. A.S. and T.R.G. acknowledge support from U54HG006093. R.M.G. acknowledges funding support from the Mystic Force Foundation. The Mayo Clinic Brain Tumor Patient-Derived Xenograft National Resource is supported by Mayo Clinic, the Mayo SPORE in Brain Cancer CA108961and NIH grant R24 NS92940. We gratefully acknowledge single cell library preparation and next generation sequencing services provide by the Onco-Genomics Shared Resource, Sylvester Comprehensive Cancer Center, University of Miami. SLW is supported by R21CA216227.

## Author contributions

V.S. conceived and designed research, performed experiments, developed methods, analyzed data, and wrote the paper. A.M.J. designed research, performed experiments, developed methods, analyzed data, and wrote the paper. M.E.M performed experiments. M.F. developed methods and analyzed data. W.W. performed experiments. R. K.S. performed experiments. M.A.D. developed methods and analyzed data. S.W. provided critical equipment. J.W.H. provided critical equipment. C-H.V. provided critical equipment and performed experiments. N.J.L. provided critical equipment and performed experiments. C.W. provided critical equipment and performed experiments. R.M.G. performed experiments. M.I. provided patient specimens. R.J.K. provided patient specimens. J.S. provided critical data and resources. A.S. provided critical data and resources. T.R.G. provided critical equipment and performed experiments. S.C.S. conceived and designed research, analyzed data, and wrote the paper. N.G.A. conceived and designed research, analyzed data, and wrote the paper.

## Additional information

**Competing interests:** The authors declare no competing interests.

