## [Peer Review File · Nature Communications]

Reviewers' comments:

Reviewer #1 (Remarks to the Author):

This work from the LINCS consortium employed TCGA and LINCS data to identify potential synergistic compounds for preclinical validation. Glioblastoma was used as a case study and its application to other cancers was discussed. The method of considering concordance with a reference signature and discordance with a disease signature to select candidates is novel. The finding of the correlation between synergy scores and discordance is very interesting. It demonstrates the promise of using LINCS data for combination discovery. The web platform and the source code could help researchers discover new combinations and reproduce their work.

The title and introduction seem to state novel therapeutic combinations for GBM were discovered in this work, but neither animal and mechanistic studies were conducted, nor the challenges of BBB and the potential clinical application were investigated. Therefore, a more appropriate title would be something like drug-disease signature integration identifies therapeutic combinations for glioblastoma preclinical studies. Having said that, I think a comparison with conventional high-throughput screening approaches for combination discovery should be discussed.

A clinically meaningful combination is probably the one combining with the existing therapy TMZ. The potential to apply this method to identify drugs combining with TMZ was mentioned at the end, but as indicated in table S1, the TMZ signature is very small. Could you justify the feasibility of this approach in such cases where the signature has little overlap with L1000 landmark genes?

TCS is computed by counting the number of cell lines where the gene is up-/down-regulated. This simple method may ignore the complexity of the biology. Studies have shown that drug profiles are highly dependent on cell lines, dosage, and treatment duration (PMIDS: 29084964, 28699633). Now LINCS profiles have covered over 70 cell lines under various biological conditions, treating profiles equally is certainly not making sense, how does the approach address this?

What's the cutoff to define up-/down- regulated genes?

It would be nice to gain some insights of the mechanism from the analysis. At least the genes/pathways contributing to the synergistic effect should be analyzed.

RNA-Seq data of TCGA and GBM tumors was described in the Method, while agilent microarray data was used for the analysis on page 10. Please clarify.

RPMK was used to measure expression in PDX GBM samples while a different quantification method was used in TCGA samples, it might not be reasonable to compute Pearson correlations among those samples directly.

TCGA includes much more number of cancer types than ten used in figure 1, please justify and add the time of collecting data from TCGA.

Figure 1: (a) (b) are mislabeled.

page 10: the clustering of GBM samples does not indicate L1000 profiles would sufficiently describe the impact of all small molecule treatments. It simply implies the landmark genes could capture the tissue of origin of these samples.

Figure 4bc and figure 5cd, how to compute discordance within each cell line? It seems a signature of each cell line was derived from base line profiles. What's the cutoff? A more detailed description is expected.

Reviewer #2 (Remarks to the Author):

This paper compares drug profiling results in GBM cultures to drug profiling results in the L1000 data, for non-GBM cell lines. They use ad hoc correlation-based scores to select possible synergistic drugs. In its present scope and form, the paper lacks majorly in development, method characterization and validation. It does not provide a novel or credible case for a method that is well suited to identify combinations against cancer. Major concerns:

- 1) An existing paper describes the use of connectivity map searches for neural cancers, as do some recently introduced software algorithms (ccmap, drugpairseeker, de Preter, clin cancer res 2009).
- 2) Other existing work has shown that screening just random drugs in GBM and other lines, can detect a substantial frequency of synergies (schmidt, neurooncology 2013, holbeck et al, cancer res 2017). This, together with the lack of systematic evaluation of multiple predictions, calls into question the sensitivity and specificity of the computational pipeline as a tool to find synergies.
- 3) The analysis of separation of tcga cancers is un-informative, a similar separation is most likely achievable by randomly selecting a few hundred gene markers, and this does not support that the the proposed method is an accurate detector of drug effects in GBM.
- 4) There is no systematic consideration of normalization/data quality problems in the L1000, which is a hot discussion point in the field. The analysis uses a small set of compounds compared to the current connectivity map, a broader analysis would be needed to characterize the pipeline.
- 5) Existing work develops computational techniques to define combinations from data integration. The paper suggests an ad hoc scoring system considering the already existing frameworks for searching compounds in L1000, without consideration of obvious alternatives available (e.g. adaptation of the existing GSEA/connectivity map search algorithm, or above packages).
- 6) I did not find any convincing in vivo validation of the finding, which together with the lack of theoretical justification and systematic empirical validation precludes publication in a upper tier journal.

Reviewer #3 (Remarks to the Author):

In this paper, the authors developed a novel analysis pipeline that integrates glioblastoma tumor transcriptome data with transcriptome signatures of xenograft cell lines in response to chemotherapy and infers synergistic drug combinations. The pipeline begins with a 978-gene expression signature that was shown to be capable of stratifying tumor samples by cancer type, and was used to measure response of cancer cell lines to drug perturbations in the LINCS project. Using this gene set, the authors experimentally determined the expression signature of GBM cell lines in response to JQ1, a BET inhibitor that may overcome the commonly observed temozolomide resistance in GBM. Cross referencing existing signature genes common to all other cell lines ('transcriptional consensus signature', or TCS) for each compound, the authors then demonstrate that the GBM-JQ1 signature is concordant with the overall JQ1 signature in other cell lines, with a high overlap in the gene sets as well as consistency in the directionality of changes in expression. Having confirmed that the TCSs for the drug compendium used in the LINCS project could cluster the drugs by their mechanisms of action, the paper puts forward the hypothesis that drugs which drive expression in a direction that is opposite to the changes from normal to cancerous tissues and shows a signature gene set orthogonal to that of JQ1 should generate synergistic effects when dosed in combination with JQ1. To test this hypothesis, the authors score each compound using a metric that rewards discordance with both the disease expression signature and the JQ1 signature in GBM. This analysis revealed GSK-1070916, an Aurora B kinase inhibitor, as a top scoring drug.

Its synergy with GBM is then experimentally verified using combination dosing. Finally, the authors further validate the ability of the pipeline to uncover synergistic drug pairs by showing that mitoxantrone and imatinib, two drugs with top orthogonality scores to gemcitabine, show significant synergy with gemcitabine in killing GBM cells.

The methodology described in this paper seems to be widely applicable to various types of cancer where multiple patient-derived cell lines and transcriptome profiles are available. Since the pipeline allows prediction of drug synergy for each single cell line, it is likely to facilitate design of personalized therapeutic strategies in GBM and other cancer types. However, to ensure rigor in the methodology and improve the fluidity of the paper, the authors need to address a few issues listed below.

1. The paper would benefit from a schematic of the overall workflow.

2. Results section 2: the authors make a connection between stratification of TCGA cancer types using baseline expression signatures and the utility of the same signature gene set to infer drug response. The inherent logic here seems elusive, since genes that confer tissue-specificity in untreated cancer samples do not necessarily discriminate between the mechanisms of action of chemotherapeutic compounds.

3. Results section 3: the authors introduce the concept of Transcriptional Activity Scores (TAS) and it seemed that the purpose is to attribute the number of genes contained in the TCS of each compound to their 'pharmacological nature'. However, the correlation between TAS and the size of TCS for a given compound seems trivial, and it is unclear how such a correlation is reflective of the pharmacological properties of the compound.

4. Results section 3: to derive the GBM-JQ1 signature the authors used genes whose changes are consistent across most if not all six GBM cell lines used in the study. It is known that GBM is a highly heterogeneous disease and can at least be stratified into distinct transcriptional subtypes. What subtypes do the cell lines belong to / resemble? The authors will need to comment on the likelihood of such a consensus method potentially missing subtype-specific treatment strategies which might be key to individualized therapy for GBM.

5. Results section 5: the authors claim that the 'ideal' compound would fully reverse the 223-gene signature identified from a differential expression analysis between TCGA GBM samples and normal tissue controls. However, such a signature is only a subset of a bigger gene set used to stratify different cancer types – would the reversal of the expression of these genes be sufficient to steer the transcriptome landscape of tumor cells toward that of normal tissue cells? The authors also referred to Figure 1d for this part of the analysis, which does not appear to show any demonstration of / support for their claim.

6. To test the compounds which were inferred to be synergistic to JQ1 in GBM, the authors pick a top scoring compound, GSK-1070916, to perform combination dosing on GBM cells. The authors also include a compound with a low orthogonality score, SR1277, as negative control. While the experimental results are supportive of their predicted synergy levels, to rule out the possibility of coincidence it is useful to include another compound with an intermediate to high orthogonality score and assess if synergy with JQ1 is also concordant with predicted levels.

7. In Figure 5g the authors attempt to demonstrate a correlation among cell line discordance, Loewe combination index and orthogonality score. However, two data points is insufficient to establish convincing correlations. Availability of additional compound data would potentially strengthen the authors' argument.

8. Formatting / typographical issues:

RPMK – RPKM

'5 top / bottom genes' – actually 2 were shown

'six GBM cells' – intended to mean 'six GBM cell lines'

We thank the reviewers for their thoughtful insights and comments. We have now substantially revised our manuscript to include *in vivo* data demonstrating that aurora kinase inhibition along with bromodomain inhibition synergize to reduce tumor growth. In addition to GSK-1070916, we now demonstrate that another Aurora kinase inhibitor, alisertib, also synergizes with JQ1 *in vitro*. We utilized alisertib as it is in clinical trials for multiple cancers and is more clinically relevant than GSK-1070916. We demonstrate that combining JQ1 along with alisertib reduces tumor growth relative to either treatment alone (new Figure 6). In addition, we demonstrate that the combination is not toxic to animals over the treatment period as their weights were unaffected. These studies provide substantial *in vivo* validation for our *in vitro* studies. We would like to point out that prior to our study there were no data in LINCS from GBM or any brain tumor lines. In addition, there were no LINCS data from PDX samples from GBM lines. This is critical given findings in the field that cell lines do not recapitulate the original tumor. In fact, several studies have shown that the widely-used U87MG line is not what it was originally thought. Therefore, our studies are timely and important as they provide the scientific community with a baseline signature for GBM PDX lines. We have recently performed single cell sequencing of GBM tumors and find that there are multiple subtypes within GBM tumors as has been reported by other groups. We find that the genes responsible for the signatures for JQ1 and GSK-1070916 localize to the same cells within the tumor (Supplemental Figure 12). Therefore, we hypothesize that the synergy we are observing *in vitro* and *in vivo* is due to the compounds targeting different pathways within the same cell population rather than different parts of the tumor. We now mention this in the Discussion. To address whether it is possible to identify compounds that synergize by randomly selecting two compounds as Reviewer 2 has suggested we analyzed the NCI ALMANAC dataset and can demonstrate that compounds that synergize are more likely to be discordant to the disease signature. These findings show that not all compound combinations that are randomly selected yield synergy. We have included these new data in Supplemental Figures 7-11.

We address the reviewers' concerns below. We hope our manuscript is now acceptable for publication in *Nature Communications*.

Reviewers' comments:

Reviewer #1 (Remarks to the Author):

This work from the LINCS consortium employed TCGA and LINCS data to identify potential synergistic compounds for preclinical validation. Glioblastoma was used as a case study and its application to other cancers was discussed. The method of considering concordance with a reference signature and discordance with a disease signature to select candidates is novel. The finding of the correlation between synergy scores and discordance is very interesting. It demonstrates the promise of using LINCS data for

combination discovery. The web platform and the source code could help researchers discover new combinations and reproduce their work.

The title and introduction seem to state novel therapeutic combinations for GBM were discovered in this work, but neither animal and mechanistic studies were conducted, nor the challenges of BBB and the potential clinical application were investigated. Therefore, a more appropriate title would be something like drug-disease signature integration identifies therapeutic combinations for glioblastoma preclinical studies. Having said that, I think a comparison with conventional high-throughput screening approaches for combination discovery should be discussed.

-We thank the reviewer for their positive assessment of our manuscript. We have now included *in vivo* data demonstrating that our inhibition of Aurora kinase along with bromodomain inhibition is more effective at reducing tumor burden than either treatment alone (new Figure 6). In addition, we show in Supplemental File 6, bromodomain inhibitor combinations predicted by Synergyseq in colorectal cancer yielded MEK inhibitors. These combinations have been shown to effective *in vitro* and *in vivo*^{1,2}. We have included a comparison between our platform and NCI ALMANAC, a large pairwise-combination screen of 104 FDA approved compounds (Supplemental Files 7-11). Most importantly, high throughput screening approaches do not take into account the disease signature of each tumor, which we find is necessary for predicting synergy *in vitro* and *in vivo*.

A clinically meaningful combination is probably the one combining with the existing therapy TMZ. The potential to apply this method to identify drugs combing with TMZ was mentioned at the end, but as indicated in table S1, the TMZ signature is very small. Could you justify the feasibility of this approach in such cases where the signature has little overlap with L1000 landmark genes?

The reviewer makes an excellent point. We do find that the TMZ signature is small. We now mention the limitation of our approach for those compounds that have a limited effect in changing the L1000 genes in our Discussion. We would like to point out that therapeutically TMZ is ineffective in recurrent settings and therefore, it might be more advantageous to look for other compounds that have much more pronounced effect on L1000 genes.

TCS is computed by counting the number of cell lines where the gene is up-/down-regulated. This simple method may ignore the complexity of the biology. Studies have shown that drug profiles are highly dependent on cell lines, dosage, and treatment duration (PMIDS: 29084964, 28699633). Now LINCS profiles have covered over 70 cell lines under various biological conditions, treating profiles equally is certainly not making sense, how does the approach address this?

The reviewer makes an excellent point. This is a limitation of our current approach that we mention in our discussion. However, despite these limitations we did identify novel bromodomain

inhibitor/aurora kinase inhibitor combinations and re-analyzed other the NCI AIMANAC to show that synergy correlated with disease discordance.

What's the cutoff to define up-/down- regulated genes?

We now include the cutoff for the up- and down-regulated genes in the Methods section. Each gene of the consensus signature should be consistently up/ down regulated in at least 30% of the cell lines where the compound was tested.

It would be nice to gain some insights of the mechanism from the analysis. At least the genes/pathways contributing to the synergistic effect should be analyzed.

We have performed single cell RNA-seq analysis of GBM tumors (new Supplemental File S12) and can demonstrate that the same cell subpopulations that have the GBM signature also have the JQ1, GSK-1070916, and alisertib signatures. This suggests that synergy is occurring in the same cells. We have attempted to look at individual genes that contribute to synergy in tumors using real-time PCR. However, we could not identify a set of genes that are correlated with synergy. We suggest that single cell sequencing of treated cells will be necessary to identify genes contributing to synergy. We now mention this in the Discussion section. Moreover, we performed a functional annotation analysis on the Glioblastoma JQ1 and the GSK-1070916 TCSs using DAVID and showed that they are associated with different biological processes.

RNA-Seq data of TCGA and GBM tumors was described in the Method, while agilent microarray data was used for the analysis on page 10. Please clarify.
We have now fixed this typographical error.

RPMK was used to measure expression in PDX GBM samples while a different quantification method was used in TCGA samples, it might not be reasonable to compute Pearson correlations among those samples directly.

We have now performed Spearman correlation and found a similar effect. We have now updated the figure (modified Figure 2d).

TCGA includes much more number of cancer types than ten used in figure 1, please justify and add the time of collecting data from TCGA.

We have used all cancer types for which matched controls (normal tissue) were available at the time. The rationale for this choice, was to be able to produce disease signatures for those cancer types by comparing the tumor with the normal samples. We now mention this in the Figure legend.

Figure 1: (a) (b) are mislabeled.

We thank the reviewer for pointing this out and have modified the text accordingly.

page 10: the clustering of GBM samples does not indicate L1000 profiles would sufficiently describe the impact of all small molecule treatments. It simply implies the landmark genes could capture the tissue of origin of these samples.

The reviewer is indeed correct. We have modified the text accordingly.

Figure 4bc and figure 5cd, how to compute discordance within each cell line? It seems a signature of each cell line was derived from base line profiles. What's the cutoff? A more detailed description is expected.

We have now included a detailed description in the methods section.

Reviewer #2 (Remarks to the Author):

This paper compares drug profiling results in GBM cultures to drug profiling results in the L1000 data, for non-GBM cell lines. They use ad hoc correlation-based scores to select possible synergistic drugs. In its present scope and form, the paper lacks majorly in development, method characterization and validation. It does not provide a novel or credible case for a method that is well suited to identify combinations against cancer. Major concerns:

1) An existing paper describes the use of connectivity map searches for neural cancers, as do some recently introduced software algorithms (ccmap, drugpairseeker, de Preter, clin cancer res 2009).

We thank the reviewer for their comments. An important difference of our method with other methods is the calculation of the Transcriptional Consensus Signatures (TCS) for each compound. This is particularly relevant, because there were no GBM or any brain tumor lines used in LINCS L1000. Our platform integrates glioblastoma tumor transcriptome data with transcriptome signatures of xenograft cell lines. Such integration has not been done for L1000 cell line data and is in fact required to infer synergistic drug combinations. There are no L1000 signatures from PDX samples in LINCS. Integrating with PDX data is critical given findings in the field that cell lines do not recapitulate the original tumor. In fact, several studies have shown that the widely-used U87MG line is not what it was originally thought. We also show in Supplemental file 6 that predictions using our pipeline identify compound combinations that have been previously validated in other cancers such as BET inhibitor/MEK inhibitor combinations in colorectal cancer. Also, palbociclib combination with JQ1 has been reported⁴ and is predicted via Synergyseq. Another important difference between our method relative to other approaches is that we use a quantitative gene signature that is needed to compute disease discordance and drug concordance.

Regarding the ccmap package (released in Bioconductor in 2016), we were reluctant to utilize it as no follow-up studies to this manuscript have been published. Drugpairseeker is a useful tool, but was not appropriate for our analysis as it used a heuristic search algorithm that does not evaluate the space of all pairwise drug combinations and thus limits the number of combinations that are could potentially be predicted (the software outputs 10 combinations). Moreover, the algorithm treats each experiment (drug-cell-dose-time) individually and thus most of the predicted combinations are biased toward only a few combinations. For example, after running Drugpairseeker using our Glioblastoma Disease Signature, 9 out the 10 predicted combinations were of trichostatin A with methotrexate (multiple signatures were due to multiple cellular contexts) and the last combination was methotrexate and vorinostat.

In de Preter et al. 2016, the authors used an enrichment approach on the Connectivity Map data to identify individual drugs that would reverse their neuroblastoma signatures. This approach, however, lacks the ability to be utilized for the identification of compound combinations that take into account the overlap of their individual transcriptional responses compared to the disease signature.

2) Other existing work has shown that screening just random drugs in GBM and other lines, can detect a substantial frequency of synergies (schmidt, neurooncology 2013, holbeck et al, cancer res 2017). This, together with the lack of systematic evaluation of multiple predictions, calls into question the sensitivity and specificity of the computational pipeline as a tool to find synergies.

I think the reviewer would agree that screening just random drugs has not been overly successful in GBM. In fact, we screened many FDA approved compounds that were not effective in our PDX

lines. As shown by two large scale combination screens^{5,6} and one GBM-specific combination screen⁸, the synergy scores for each dataset follow the normal distribution and is centered at zero. Therefore, there is no bias towards having positive synergy scores by using random drugs. To address this issue, we include Figures S7-S11 to demonstrate that screening random drugs does not detect substantial frequency of synergies. Moreover we evaluated the predicted scores of our pipeline by using the NCI ALMANAC dataset.

More specifically, we evaluated our pipeline by comparing it to two combination oncology screens (the NCI ALMANAC Dataset⁵ and the Oneil et al Dataset⁶). The NCI ALMANAC Dataset contains a systematic screen of over 5000 pair-wise combinations using 104 FDA approved drugs in 60 cell lines (NCI60). In the Oneil Dataset, 38 compounds were used in 583 pairwise combinations to treat 39 cancer cell lines. To measure the combination benefit, we used the reported “ComboScore” for the NCI ALMANAC Dataset and the Lowe Additivity Score for the Oneil Dataset. Comboscores are available through the NCI Almanac data portal (<https://dtp.cancer.gov/ncialmanac/>) and the Lowe Additivity Score was obtained from a previous publication⁷. As previously reported in small scale combination screenings⁸, the distribution of synergy scored follows the normal distribution, centering around 0 (Figure S9). To adjust for the use of different synergy metrics between the two datasets, we calculated the z-scores from the two metrics.

Since the reproducibility across different single-dose datasets has been a widely acknowledged issue⁹⁻¹⁴, we proceeded with evaluating the consistency between the NCI ALMANAC and the Oneil dataset. For this we identified all the conditions (e.g. DrugA, DrugB, Cell Line) that were identical in both datasets and plotted their corresponding combination scores and z-scores (Fig S7-S11) (Pearson correlation: 0.1605269, Spearman correlation: 0.1121879).

However, given that there was no observable correlation between the two datasets, we proceeded with using the larger dataset from the NCI ALMANAC.

To examine the extent that a combination is effective across all cell lines of the same cancer type, we plotted the Comboscores of each cell line that was treated by a particular combination. As we can see in Figure S11, there is a large portion of drug combinations that exhibit high ComboScores in all cell lines belonging to the same cancer type.

Finally, we wanted to validate the synergistic potential of drug combinations that have a high Discordance Ratio, by using the NCI ALMANAC Dataset. Similar to the methodology above, we filtered for compounds that had a consensus signature of at least 40 genes that yielded a list of 657 LINCS compounds. We also filtered clinically relevant drugs that showed at least 20% decrease in Cell Growth (PercentGrowth <80) when used as single-agent treatments of cell lines. We identified 19 compounds that were used in pair-wise combinations in 33 cancer cell lines in the NCI ALMANAC Dataset. Although all combinations with a ComboScore above 0 are technically synergistic, we followed a more conservative approach, similar to previous methods⁷,

where we labeled as synergistic, the combinations with the top-ranked ComboScore values (z-score > 2.5) and as Negative, all the antagonistic, additive and low synergistic drug combinations. In Figure S11 we can see that drug combinations labeled as synergistic show a significantly higher Discordance Ratio (Wilcoxon rank sum test p-value = 2.547e-07), compared to combinations that are labeled as negative. Moreover, in Figure S9, we can see a cancer type-dependent distribution of the Discordance Ratio, where most cancer types, synergistic combinations show a higher Discordance Ratio than negative combinations. Moreover, we assessed the effect of the z-score threshold when discriminating between synergistic and negative combinations, by evaluating the median of the corresponding groups using different thresholds, indicating that regardless of the z-score threshold, Synergistic combinations have a significantly higher Discordance Score than the Negative combinations.

3) The analysis of separation of tcga cancers is un-informative, a similar separation is most likely achievable by randomly selecting a few hundred gene markers, and this does not support that the the proposed method is an accurate detector of drug effects in GBM.

We thank the reviewer for the comment. We have now rephrased that description and explained that the clustering simply shows that the L1000 genes are sufficient to distinguish between samples originating from different cancer types.

The published results suggest that there are cancer specific combinations with BET inhibitors. For instance, MEK inhibitors have been described with BET inhibitors in colorectal cancer. We do acknowledge that some compound combinations will be applicable to all cancers. However, new Figure S7-S11 using the NCI ALMANAC dataset show that L1000-based disease discordance correlates with synergy.

4) There is no systematic consideration of normalization/data quality problems in the L1000, which is a hot discussion point in the field. The analysis uses a small set of compounds compared to the current connectivity map, a broader analysis would be needed to characterize the pipeline.

We thank the reviewer for the comment. We now use the latest LINCS L1000 Dataset (March 06, 2017) that is included in GEO (GSE70138).

We refer the reviewer to our recently published manuscript that addresses this issue¹⁵.

5) Existing work develops computational techniques to define combinations from data integration. The paper suggests an ad hoc scoring system considering the already existing frameworks for searching compounds in L1000, without consideration of obvious alternatives available (e.g. adaptation of the existing GSEA/connectivity map search algorithm, or above packages).

An important difference to other methods is the calculation of the Transcriptional Consensus Signatures (TCS) for each compound. This is particularly relevant, because there were no GBM or any brain tumor lines used in LINCS L1000. In contrast to other algorithms, our method uses a quantitative gene signature that is needed to compute disease discordance and drug concordance. For example, a standard CMap search could not be used for drug synergy, because

it does not provide the specific discordant or concordant genes. In addition, our platform integrates glioblastoma tumor transcriptome data with transcriptome signatures of xenograft cell lines. There are no GBM PDX lines in publicly available databases to our knowledge. Our results showed that the PDX signature is indeed required to infer synergistic drug combinations. This is not unexpected as GBM cell lines do not recapitulate the original tumor.

6) I did not find any convincing *in vivo* validation of the finding, which together with the lack of theoretical justification and systematic empirical validation precludes publication in a upper tier journal.

We have now included *in vivo* data (Figure 6 and Supplemental File 4) demonstrating that a JQ1/alisertib combination is effective *in vivo*.

Reviewer #3 (Remarks to the Author):

In this paper, the authors developed a novel analysis pipeline that integrates glioblastoma tumor transcriptome data with transcriptome signatures of xenograft cell lines in response to chemotherapy and infers synergistic drug combinations. The pipeline begins with a 978-gene expression signature that was shown to be capable of stratifying tumor samples by cancer type, and was used to measure response of cancer cell lines to drug perturbations in the LINCS project. Using this gene set, the authors experimentally determined the expression signature of GBM cell lines in response to JQ1, a BET inhibitor that may overcome the commonly observed temozolomide resistance in GBM. Cross referencing existing signature genes common to all other cell lines ('transcriptional consensus signature', or TCS) for each compound, the authors then demonstrate that the GBM-JQ1 signature is concordant with the overall JQ1 signature in other cell lines, with a high overlap in the gene sets as well as

consistency in the directionality of changes in expression. Having confirmed that the TCSs for the drug compendium used in the LINCS project could cluster the drugs by their mechanisms of action, the paper puts forward the hypothesis that drugs which drive expression in a direction that is opposite to the changes from normal to cancerous tissues and shows a signature gene set orthogonal to that of JQ1 should generate synergistic effects when dosed in combination with JQ1. To test this hypothesis, the authors score each compound using a metric that rewards discordance with both the disease expression signature and the JQ1 signature in GBM. This analysis revealed GSK-1070916, an Aurora B kinase inhibitor, as a top scoring drug. Its synergy with GBM is then experimentally verified using combination dosing. Finally, the authors further validate the ability of the pipeline to uncover synergistic drug pairs by showing that mitoxantrone and imatinib, two drugs with top orthogonality

scores to gemcitabine, show significant synergy with gemcitabine in killing GBM cells.

The methodology described in this paper seems to be widely applicable to various types of cancer where multiple patient-derived cell lines and transcriptome profiles are available. Since the pipeline allows prediction of drug synergy for each single cell line, it is likely to facilitate design of personalized therapeutic strategies in GBM and other cancer types. However, to ensure rigor in the methodology and improve the fluidity of the paper, the authors need to address a few issues listed below.

1. The paper would benefit from a schematic of the overall workflow.

We thank the reviewer for their positive assessment of our manuscript. We have now included a workflow as suggested by the reviewer (new Figure 1).

2. Results section 2: the authors make a connection between stratification of TCGA cancer types using baseline expression signatures and the utility of the same signature gene set to infer drug response. The inherent logic here seems elusive, since genes that confer tissue-specificity in untreated cancer samples do not necessarily discriminate between the mechanisms of action of chemotherapeutic compounds.

We thank the reviewer for bringing up this point, which needs further explanation. The main point here is to demonstrate that the L1000 gene set is sufficient to characterize the cancer and therefore should also be sufficient to characterize specific perturbations.

This is now explained better. The 978 genes that are measured by the L1000 platform, are the result of a dimension reduction procedure on 12,063 highly diverse Affymetrix gene expression profiles extracted from over 400+ GEO Series that reflect both tissue-specific transcriptional changes and perturbation-induced transcriptional changes.

The platform relies on both the baseline disease signature and the signature from the compounds used. Prior to our study there were no data in LINCS from GBM or any brain tumor lines and there was no data PDX samples. The integration of these data with LINCS via the TCS is important given findings in the field that cell lines do not recapitulate the original tumor.

In fact, several studies have shown that the widely-used U87MG line is not what it was originally thought. The TCS based on L1000 enable this integration.

3. Results section 3: the authors introduce the concept of Transcriptional Activity Scores (TAS) and it seemed that the purpose is to attribute the number of genes contained in the TCS of each compound to their 'pharmacological nature'. However, the correlation between TAS and the size of TCS for a given compound seems trivial, and it is unclear how such a correlation is reflective of the pharmacological properties of the compound.

We thank the reviewer for bringing up this important point. Yes that is correct. We were merely trying to explain the variability in the number of TCS genes that various drugs have, by comparing it to BROAD's QC metric (TAS). This is now explained better in the manuscript.

4. Results section 3: to derive the GBM-JQ1 signature the authors used genes whose changes are consistent across most if not all six GBM cell lines used in the study. It is known that GBM is a highly heterogeneous disease and can at least be stratified into distinct transcriptional subtypes. What subtypes do the cell lines belong to / resemble? The authors will need to comment on the likelihood of such a consensus method potentially missing subtype-specific treatment strategies which might be key to individualized therapy for GBM.

This is indeed a critical point. We have now commented on the likelihood of our approach identifying combinations for each GBM subtype. We have recently performed single cell sequencing of GBM tumors and find that there are multiple subtypes within the tumor and now included these results into the revised manuscript. (Figure S12). We hypothesize that the synergy we are observing *in vitro* and *in vivo* is due to the compounds targeting different pathways within the same cell population rather than different parts of the tumor. We now explain this in the Discussion.

5. Results section 5: the authors claim that the ‘ideal’ compound would fully reverse the 223-gene signature identified from a differential expression analysis between TCGA GBM samples and normal tissue controls. However, such a signature is only a subset of a bigger gene set used to stratify different cancer types – would the reversal of the expression of these genes be sufficient to steer the transcriptome landscape of tumor cells toward that of normal tissue cells? The authors also referred to Figure 1d for this part of the analysis, which does not appear to show any demonstration of / support for their claim.

We understand the potential limitations of only using the 978-gene signature of the L1000 platform. Ideally, our approach would be improved using RNA-seq data, but measuring over a million perturbation responses using RNA-seq would be logistically and financially difficult and such a dataset does not exist. The L1000 landmark genes were selected by a comprehensive data reduction process with the specific intent to represent the entire genome and thus would be sufficient as transcriptional markers and capture the necessary changes that would steer the cancer transcriptome landscape towards the normal one. The utility of the L1000 gene set to differentiate cancer types and mechanism of action of small molecule perturbagens are demonstrated in figures 2 and 4, respectively. We are therefore confident that the L1000 gene set is sufficient and that the L1000 dataset is useful; although a more comprehensive coverage would almost certainly be better.

6. To test the compounds which were inferred to be synergistic to JQ1 in GBM, the authors pick a top scoring compound, GSK-1070916, to perform combination dosing on GBM cells. The authors also include a compound with a low orthogonality score, SR1277, as negative control. While the experimental results are supportive of their predicted synergy levels, to rule out the possibility of coincidence it is useful to include another compound with an intermediate to high orthogonality score and assess if synergy with JQ1 is also concordant with predicted levels.

We thank the reviewer for the suggestion. We have now shown that alisertib (a high scoring compound) also synergizes with JQ1. Our SynergySeq platform predicts that palbociclib should also synergize with JQ1. Indeed, recent studies have shown that this is the case⁴. This synergy has actually been reported in other cancers and therefore attests to the validity of our platform.

7. In Figure 5g the authors attempt to demonstrate a correlation among cell line discordance, Loewe combination index and orthogonality score. However, two data points is insufficient to establish convincing correlations. Availability of additional compound data would potentially strengthen the authors' argument.

We thank the reviewer for pointing this out and have updated our figure and manuscript. We now show that a compound that is not predicted to synergize with JQ1, SR1277 does not synergize. We have also demonstrated that JQ1 synergizes with GSK1070916 and now added a second aurora kinase inhibitor alisertib. We also performed an FDA approved compound screen in PDX lines and demonstrate that synergy can be predicted of active compounds.

8. Formatting / typographical issues:

RPMK – RPKM

We have modified the text accordingly.

'5 top / bottom genes' – actually 2 were shown

'six GBM cells' – intended to mean 'six GBM cell lines'

We have modified the text accordingly.

References

- 1 Wyce, A. *et al.* MEK inhibitors overcome resistance to BET inhibition across a number of solid and hematologic cancers. *Oncogenesis* **7**, 35, doi:10.1038/s41389-018-0043-9 (2018).
- 2 Ma, Y. *et al.* The MAPK Pathway Regulates Intrinsic Resistance to BET Inhibitors in Colorectal Cancer. *Clin Cancer Res* **23**, 2027-2037, doi:10.1158/1078-0432.CCR-16-0453 (2017).
- 3 Subramanian, A. *et al.* A Next Generation Connectivity Map: L1000 Platform and the First 1,000,000 Profiles. *Cell* **171**, 1437-1452 e1417, doi:10.1016/j.cell.2017.10.049 (2017).
- 4 Sun, B. *et al.* Synergistic activity of BET protein antagonist-based combinations in mantle cell lymphoma cells sensitive or resistant to ibrutinib. *Blood* **126**, 1565-1574, doi:10.1182/blood-2015-04-639542 (2015).
- 5 Holbeck, S. L. *et al.* The National Cancer Institute ALMANAC: A Comprehensive Screening Resource for the Detection of Anticancer Drug Pairs with Enhanced Therapeutic Activity. *Cancer research* **77**, 3564-3576, doi:10.1158/0008-5472.CAN-17-0489 (2017).
- 6 O'Neil, J. *et al.* An Unbiased Oncology Compound Screen to Identify Novel Combination Strategies. *Mol Cancer Ther* **15**, 1155-1162, doi:10.1158/1535-7163.MCT-15-0843 (2016).

- 7 Preuer, K. *et al.* DeepSynergy: predicting anti-cancer drug synergy with Deep Learning. *Bioinformatics* **34**, 1538-1546, doi:10.1093/bioinformatics/btx806 (2018).
- 8 Schmidt, L. *et al.* Comparative drug pair screening across multiple glioblastoma cell lines reveals novel drug-drug interactions. *Neuro-oncology* **15**, 1469-1478, doi:10.1093/neuonc/not111 (2013).
- 9 Mpindi, J. P. *et al.* Consistency in drug response profiling. *Nature* **540**, E5-E6, doi:10.1038/nature20171 (2016).
- 10 Haverty, P. M. *et al.* Reproducible pharmacogenomic profiling of cancer cell line panels. *Nature* **533**, 333-337, doi:10.1038/nature17987 (2016).
- 11 Haibe-Kains, B. *et al.* Inconsistency in large pharmacogenomic studies. *Nature* **504**, 389-393, doi:10.1038/nature12831 (2013).
- 12 Safikhani, Z. *et al.* Safikhani et al. reply. *Nature* **540**, E11-E12, doi:10.1038/nature20581 (2016).
- 13 Bouhaddou, M. *et al.* Drug response consistency in CCLE and CGP. *Nature* **540**, E9-E10, doi:10.1038/nature20580 (2016).
- 14 Weinstein, J. N. & Lorenzi, P. L. Cancer: Discrepancies in drug sensitivity. *Nature* **504**, 381-383, doi:10.1038/nature12839 (2013).
- 15 Stathias, V. *et al.* Sustainable data and metadata management at the BD2K-LINCS Data Coordination and Integration Center. *Sci Data* **5**, 180117, doi:10.1038/sdata.2018.117 (2018).

Reviewers' comments:

Reviewer #1 (Remarks to the Author):

The authors have addressed the majority of my comments. They added in vivo validation data, however, they validated the alisertib and JQ-1 combination instead of the GSK-1070916 and JQ-1 combination. They justified this by stating that because alisertib is in clinical trials for multiple cancers and is predicted to have better PK properties than GSK-1070916. I am very surprised by this justification. First, GSK-1070916 is a potent and selective ATP-competitive inhibitor of aurora B and aurora C with K_{is} of 0.38 and 1.5 nM, respectively, and is >250- fold selective over Aurora A (taken from medchemexpress), while alisertib is highly selective small molecule inhibitor of the serine/threonine protein kinase Aurora A (taken from NCI). Even though both are aurora kinase inhibitors, they have different MOA. Moreover, they failed to demonstrate that both inhibitors share similar TCS, which means that alisertib should be predicted to synergize with JQ1 before showing any synergistic effect in vitro. I am also surprised that neither alisertib nor JQ1 alone exhibits the anti-cancer effect in vivo because they are discordant to GBM DEGs.

Regarding the comparison with screening data, Fig S10 is confusing, with little details, it is hard to tell their method is better than random selection.

TCS is computed by the summarization of all profiles without considering the confounding effect of treatment conditions. The profiles for the same compound could be dramatically different between 10um and 5um. It would be fine if the limitation were thoroughly discussed, but I did not see any related papers were cited nor the limitation was highlighted.

The legend of Fig S4 is poorly described.

Reviewer #2 (Remarks to the Author):

I appreciated the addition of an in vivo experiment to support the claim of 'therapeutic' combinations and the tendency towards method evaluation.

I am still struck by the discrepancy between claims made in the title, abstract and text and the actual content of the paper. To say that a method identifies therapeutic combinations is over-claim for three reasons. First, 'identify' would imply truly excellent sensitivity and specificity of a method that classifies synergies from non-synergies. I do not see that this is addressed by the paper. Second, while the paper now contains some comparisons to foreseeable alternative ways of performing the calculation, the paper basically says that it is thanks to this particular computational procedure the combinations are found. The conceptual or practical value of the particular computational methods used remains unaddressed in the main text of the paper, although there is some useful empirical studies in the Supplement. Thirdly, to claim therapeutic success is a slippery slope despite mouse experiments; There are (literally) dozens of papers that record similar therapeutic success against GBM in mice. Lastly, I noted that the abstract and introduction say that prevention of drug resistance is a key impetus for this work. This is also not shown by the paper.

In my view, for this study to be acceptable in any scientific journal, the authors must nuance the claims made to a degree that is consistent with good scientific practice, and signal more clearly an awareness of the ad hoc nature of the approach (or demonstrate its superiority). This is a nice study that uses connectivity map and other data to find compounds that synergize with JQ1, which were validated in vivo. Many readers would appreciate it as such, without the hyperbole.

Reviewer #3 (Remarks to the Author):

I am satisfied with the changes made in the revised manuscript and support publication of this work.

Reviewer #1 (Remarks to the Author):

The authors have addressed the majority of my comments. They added in vivo validation data, however, they validated the alisertib and JQ-1 combination instead of the GSK-1070916 and JQ-1 combination. They justified this by stating that because alisertib is in clinical trials for multiple cancers and is predicted to have better PK properties than GSK-1070916. I am very surprised by this justification. First, GSK-1070916 is a potent and selective ATP-competitive inhibitor of aurora B and aurora C with K_i of 0.38 and 1.5 nM, respectively, and is >250- fold selective over Aurora A (taken from medchemexpress), while alisertib is highly selective small molecule inhibitor of the serine/threonine protein kinase Aurora A (taken from NCI). Even though both are aurora kinase inhibitors, they have different MOA. Moreover, they failed to demonstrate that both inhibitors share similar TCS, which means that alisertib should be predicted to synergize with JQ1 before showing any synergistic effect in vitro. I am also surprised that neither alisertib nor JQ1 alone exhibits the anti-cancer effect in vivo because they are discordant to GBM DEGs.

Regarding the comparison with screening data, Fig S10 is confusing, with little details, it is hard to tell their method is better than random selection.

-We have now removed this figure as it was confusing to the reviewer.

TCS is computed by the summarization of all profiles without considering the confounding effect of treatment conditions. The profiles for the same compound could be dramatically different between 10um and 5um. It would be fine if the limitation were thoroughly discussed, but I did not see any related papers were cited nor the limitation was highlighted.

-We have now discussed this limitation in the Discussion.

The legend of Fig S4 is poorly described.

-We have now replaced Table in Figure S4 for two new figures that demonstrate that Alisertib and GSK-1070916 have similar Transcriptional signatures.

We thank the reviewer for the positive assessment of our manuscript. We now demonstrate that GSK-1070916 and alisertib have related TCS (Supplemental file 4) We also include references that demonstrate that both compounds inhibit Aurora kinase A and B used in our in vitro and in vivo studies. Several papers and reviews¹ have shown that both alisertib and GSK-1070916 inhibit Aurora kinase A and B in the low nanomolar range in vitro in kinase assays and in the 100nM range in cell based assays¹. Since we observe synergy starting at approximately 300nM we suggest that at this concentration both Aurora kinase A and Aurora kinase B are inhibited. While Aurora kinase C may be a target in some cancers we recently did single cell

sequencing of a GBM tumor and find that while aurora kinase a and b are expressed in the neoplastic part of the tumor, aurora kinase c is sparsely expressed in the myeloid part of the tumor. We now include these new data in Supplemental Figure 11. For these reasons we suggest that Aurora kinase A and B are the relevant targets in GBM. We now mention this in the Discussion.

Regarding the reviewer being surprised that neither JQ1 nor alisertib having anti-cancer effects in vivo, we suggest that this is due to the doses used. As seen in multiple studies, anti-cancer effects with JQ1 is achieved only at certain doses²⁻⁶. However, at higher doses there is the issue of toxicity and therefore utilizing lower doses of both compounds in many instances is more desirable. In our hands, we saw no toxicity using the indicated doses of JQ1 or alisertib or the combinations we did not see a decrease in mouse weights.

Reviewer #2 (Remarks to the Author):

I appreciated the addition of an in vivo experiment to support the claim of 'therapeutic' combinations and the tendency towards method evaluation.

I am still struck by the discrepancy between claims made in the title, abstract and text and the actual content of the paper. To say that a method identifies therapeutic combinations is over-claim for three reasons. First, 'identify' would imply truly excellent sensitivity and specificity of a method that classifies synergies from non-synergies. I do not see that this is addressed by the paper. Second, while the paper now contains some comparisons to foreseeable alternative ways of performing the calculation, the paper basically says that it is thanks to this particular computational procedure the combinations are found. The conceptual or practical value of the particular computational methods used remains unaddressed in the main text of the paper, although there is some useful empirical studies in the Supplement. Thirdly, to claim therapeutic success is a slippery slope despite mouse experiments; There are (literally) dozens of papers that record similar therapeutic success against GBM in mice. Lastly, I noted that the abstract and introduction say that prevention of drug resistance is a key impetus for this work. This is also not shown by the paper.

In my view, for this study to be acceptable in any scientific journal, the authors must nuance the claims made to a degree that is consistent with good scientific practice, and signal more clearly an awareness of the ad hoc nature of the approach (or demonstrate its superiority). This is a nice study that uses connectivity map and other data to find compounds that synergize with JQ1, which

were validated in vivo. Many readers would appreciate it as such, without the hyperbole.

We thank the reviewer for their thoughtful comments and our positive assessment of our work. We have now changed the word “therapeutic” in the title to “synergistic.” In addition, we have removed the word therapeutic throughout the text. Finally, we have removed “resistance” from both the introduction and discussion. We hope these modifications make the paper now acceptable for publication in “ Nature communications.”

References

- 1 de Groot, C. O. *et al.* A Cell Biologist's Field Guide to Aurora Kinase Inhibitors. *Front Oncol* **5**, 285, doi:10.3389/fonc.2015.00285 (2015).
- 2 Karakashev, S. *et al.* BET Bromodomain Inhibition Synergizes with PARP Inhibitor in Epithelial Ovarian Cancer. *Cell Rep* **21**, 3398-3405, doi:10.1016/j.celrep.2017.11.095 (2017).
- 3 Lam, L. T. *et al.* Vulnerability of Small-Cell Lung Cancer to Apoptosis Induced by the Combination of BET Bromodomain Proteins and BCL2 Inhibitors. *Mol Cancer Ther* **16**, 1511-1520, doi:10.1158/1535-7163.MCT-16-0459 (2017).
- 4 Schaffer, M. *et al.* Identification of potential ibrutinib combinations in hematological malignancies using a combination high-throughput screen. *Leuk Lymphoma* **59**, 931-940, doi:10.1080/10428194.2017.1349899 (2018).
- 5 Siegel, M. B. *et al.* Small molecule inhibitor screen identifies synergistic activity of the bromodomain inhibitor CPI203 and bortezomib in drug resistant myeloma. *Oncotarget* **6**, 18921-18932, doi:10.18632/oncotarget.4214 (2015).
- 6 Yang, L. *et al.* Repression of BET activity sensitizes homologous recombination-proficient cancers to PARP inhibition. *Sci Transl Med* **9**, doi:10.1126/scitranslmed.aal1645 (2017).

REVIEWERS' COMMENTS:

Reviewer #1 (Remarks to the Author):

My comments were addressed. Thanks.